# Septins function in exocytosis via physical interactions with the exocyst complex in fission yeast cytokinesis

**Davinder Singh[1][†], Yajun Liu[1][†], Yi-Hua Zhu[1], Sha Zhang[1], Shelby M Naegele[1], Jian-Qiu Wu[1,2]***

[1]Department of Molecular Genetics, The Ohio State University, Columbus, United States; [2]Department of Biological Chemistry and Pharmacology, The Ohio State University, Columbus, United States

**\*For correspondence:**
wu.620@osu.edu

[†]These authors contributed equally to this work

## eLife Assessment

How secretion is regulated during cell division and how membrane trafficking factors cooperate with the cytoskeleton during cell division remain poorly understood. In this work the authors find protein-protein interactions and localization dependencies between the polymeric septin cytoskeleton and the exocyst complex, using fission yeast as a model organism and using alphafold 3 based structural predictions. The work provides a **valuable** body of new information that will be of great interest to the cell biology community. The evidence is **solid** and provides the authors and the community a framework to test if the identified interfaces reflect bona fide interaction sites in vivo and in vitro in future.

**Abstract** Septins can function as scaffolds for protein recruitment, membrane-bound diffusion barriers, or membrane curvature sensors. Septins are important for cytokinesis, but their exact roles are still obscure. In fission yeast, four septins (Spn1–Spn4) accumulate at the rim of the division plane as rings. The octameric exocyst complex, which tethers exocytic vesicles to the plasma membrane, exhibits a similar localization and is essential for plasma membrane deposition during cytokinesis. Without septins, the exocyst spreads across the division plane but is absent from the rim during septum formation. These results suggest that septins and the exocyst physically interact for proper localization and function. Indeed, we predicted six pairs of interactions between septin and exocyst subunits by AlphaFold, most of them are confirmed by co-immunoprecipitation and yeast two-hybrid assays. Exocyst mislocalization results in mistargeting of secretory vesicles and their cargos, which leads to cell-separation delay in septin mutants. Our results indicate that septins guide the targeting of the exocyst complex on the plasma membrane for vesicle tethering during cytokinesis through physical interactions.

## Introduction

Septins are a family of GTP-binding proteins that are highly conserved from yeast to mammalian cells (*Longtine et al., 1996*; *Gladfelter et al., 2001*; *Cao et al., 2007*; *Nishihama et al., 2011*; *Onishi and Pringle, 2016*; *Marquardt et al., 2019*; *Woods and Gladfelter, 2021*). They form hetero-oligomeric complexes that can assemble into different higher-order structures such as rings, gauzes, hourglasses, bars, and carry out various functions (*Frazier et al., 1998*; *Hsu et al., 1998*; *Kinoshita, 2003*; *Sheffield et al., 2003*; *Bertin et al., 2008*; *Garcia et al., 2011*; *Bridges et al., 2014*). Septins can serve as scaffolds for protein recruitment at discrete cellular locations (*Gladfelter et al., 2001*;

*Versele and Thorner, 2005*; *Mostowy and Cossart, 2012*; *Finnigan et al., 2015*; *Meitinger and Palani, 2016*; *Perez et al., 2016*; *Marquardt et al., 2019*). Septins are proposed to act as a diffusion barrier to ensure that cellular components are spatially segregated or compartmentalized (*Dobbelaere and Barral, 2004*; *Caudron and Barral, 2009*; *Hu and Nelson, 2011*; *McMurray et al., 2011*). They can also sense the membrane curvatures and/or deform the plasma membrane due to their lipid-binding properties (*Bridges and Gladfelter, 2016*; *Cannon et al., 2017*; *Cannon et al., 2019*; *McMurray, 2019*; *Shi et al., 2023*). The diverse roles of septins lead to their involvement in multiple processes including cytokinesis, mitosis, exocytosis, apoptosis, fungal or viral infections, neuronal spine morphogenesis, ciliogenesis, and spermiogenesis (*Longtine et al., 1996*; *Kartmann and Roth, 2001*; *Mostowy and Cossart, 2012*; *Dolat et al., 2014*; *Momany and Talbot, 2017*; *Ageta-Ishihara and Kinoshita, 2021*; *Neubauer and Zieger, 2021*; *Woods and Gladfelter, 2021*; *Safavian et al., 2023*).

One of the best-studied septin functions is their roles in cytokinesis in the budding yeast *Saccharomyces cerevisiae* (*Hartwell, 1971*; *Haarer and Pringle, 1987*; *Ford and Pringle, 1991*; *Kim et al., 1991*; *DeMarini et al., 1997*; *Bi et al., 1998*; *Lippincott and Li, 1998*; *Longtine et al., 1998*). The septin ring or hourglass structures at the presumptive bud site and bud neck are required for the recruitment and maintenance of various cytokinesis proteins (*Bi et al., 1998*; *Lippincott and Li, 1998*; *Gladfelter et al., 2001*; *Longtine and Bi, 2003*; *Marquardt et al., 2019*). These roles occur through either direct interactions with proteins such as the F-BAR protein Hof1 (*Vallen et al., 2000*; *Meitinger et al., 2013*; *Oh et al., 2013*), or through septin-binding proteins such as Bni5, which links septins to the myosin-II heavy chain Myo1 (*Lee et al., 2002*; *Fang et al., 2010*; *Finnigan et al., 2015*). During cytokinesis, the septin double rings were proposed to function as a diffusion barrier for proteins such as the exocyst component Sec3 and chitin synthase II (Chs2) at the division site (*Dobbelaere and Barral, 2004*). But other studies have challenged this view by showing that Chs2 localizes efficiently to the division site in the absence of septin rings (*Wloka et al., 2011*). Regardless of the debate, septins are known to play essential roles in budding yeast cytokinesis. However, no physical interactions between septins and the exocyst have been reported, even in the genome-wide interactome studies (*Michaelis et al., 2023*). Moreover, budding yeast septins also serve as scaffolds for the localization of many proteins at the bud neck including signaling proteins, bud site selection proteins, and chitin synthases (*Longtine et al., 1996*; *DeMarini et al., 1997*; *Gladfelter et al., 2001*; *Longtine and Bi, 2003*; *Marquardt et al., 2019*).

Unlike in budding yeast, septins are not essential in the fission yeast *Schizosaccharomyces pombe*, and their roles in cytokinesis remain obscure (*Longtine et al., 1996*; *Berlin et al., 2003*; *Tasto et al., 2003*; *Wu et al., 2010*; *Zheng et al., 2024*). Fission yeast has seven septins, Spn1–Spn7, with Spn1–Spn4 expressing in vegetative cells and functioning at the division site (*Longtine et al., 1996*; *Berlin et al., 2003*; *Tasto et al., 2003*; *Wu et al., 2003*; *An et al., 2004*; *Petit et al., 2005*; *Onishi et al., 2010*; *Wu et al., 2010*). None of the septins Spn1–Spn4 is essential, but loss of some or all of them causes a delay in cell separation, resulting in a multiseptated phenotype (*An et al., 2004*). Spn1 and Spn4 are the more important components of the septin ring structure during cytokinesis (*An et al., 2004*). Septins accumulate to the division site shortly before the contractile ring constriction and form a single ring which quickly transitions into unconstricting double rings (*Berlin et al., 2003*; *Tasto et al., 2003*; *Wu et al., 2003*; *Wu et al., 2010*). It is only known that septins recruit the anillin Mid2 (*Berlin et al., 2003*; *Tasto et al., 2003*), the guanine nucleotide exchange factor Gef3 (*Muñoz et al., 2014*; *Wang et al., 2015*), the small GTPase Rho4 (*Wang et al., 2015*), and the glucanases Eng1 and Agn1 to the division site (*Martín-Cuadrado et al., 2005*). Thus, we know much less about fission yeast septins compared to those in budding yeast. It remains mysterious what the most conserved functions of septins are during evolution.

Previous studies have suggested that septins function in exocytosis (*Hsu et al., 1998*; *Vega and Hsu, 2003*; *Martín-Cuadrado et al., 2005*; *Pérez et al., 2015*; *Tokhtaeva et al., 2015*). Fission yeast septins are proposed to work with the exocyst complex to regulate the secretion of glucanases at the appropriate location, but it was reported that septins and the exocyst are independent for localization (*Martín-Cuadrado et al., 2005*; *Pérez et al., 2015*). The exocyst is a highly conserved, octameric complex (Sec3, Sec5, Sec6, Sec8, Sec10, Sec15, Exo70, and Exo84) in exocytosis (*TerBush et al., 1996*; *Wang et al., 2002*; *Hsu et al., 2004*; *Heider and Munson, 2012*; *Liu and Guo, 2012*). It functions in late stages of exocytosis by promoting the tethering and fusion of post-Golgi secretory

vesicles to the plasma membrane (*TerBush et al., 1996*; *Hsu et al., 2004*; *Liu et al., 2018*). Although studies suggest that septins regulate the exocyst complex and a possible involvement of septins in targeting secretory vesicles to the exocytic sites (*Hsu et al., 1998*; *Vega and Hsu, 2003*; *Li et al., 2007*; *Gupta et al., 2015*), no direct physical interactions between septins and the exocyst subunits have been reported in budding yeast or other organisms except rat brain (*Hsu et al., 1998*).

Here we report that septins regulate the exocyst localization and vesicle targeting in fission yeast via physical interactions. We find that the loss of septin rings alters the exocyst localization, with increased concentration to the center and reduced localization to the rim of the division plane. The initial recruitment of the exocyst is independent of septins, but the exocyst requires septin rings to maintain the rim localization during furrow ingression. Consistently, we found multivalent physical interactions consistent with direct binding between septins and the exocyst subunits. Loss of the exocyst ring leads to abnormal accumulation of secretory vesicles in septin mutants. As a result, the glucan synthase Bgs1/Cps1 accumulates more to the cell center, and the glucanase Eng1 is missing from the rim of the division plane, contributing to delayed cell separation and a thicker septum in septin mutant cells. Our findings provide insights into the regulation of the exocyst localization and function on the plasma membrane by septins in other systems.

## Results
### The septin and exocyst complex colocalize and are partially interdependent for localization at the division site

Both septins and the exocyst complex localize to the division site during cytokinesis (*Longtine et al., 1996*; *Wang et al., 2002*; *An et al., 2004*; *Petit et al., 2005*). To understand if and how they work together, we first examined the colocalization of the septin Spn1 and the exocyst subunit Sec3 in fission yeast. Spn1 is a key component in septin structures, and its deletion leads to a complete loss of all septins from the division site (*An et al., 2004*). Sec3 is a spatial landmark for exocytosis in budding yeast (*Finger et al., 1998*; *Boyd et al., 2004*; *Luo et al., 2014*). The fission yeast Sec3 is an essential gene and crucial for exocyst localization (*Kim et al., 2010*; *Bendezú et al., 2012*; *Jourdain et al., 2012*). Spn1 and Sec3 colocalized at the division site as a single ring first, and later as double rings during septum formation (*Figure 1A, B*). The colocalization of Spn1 with another exocyst subunit Exo70 was confirmed using SoRa (Super resolution by Optical Re-Assignment) spinning disk confocal microscopy (*Figure 1C*). Sec3 and Exo70, but not Spn1, also concentrated at cell tips (*Figure 1C, D*). However, Sec3 arrived at the division site 13.4 ± 2.2 min after spindle pole body separation, about 10 min earlier than Spn1, which arrived at 23.6 ± 1.8 min (*Figure 1D, E*). Time-lapse movies of Exo70-tdTomato and Spn1-mEGFP confirmed that the exocyst appeared at the division site earlier than septins (*Figure 1—video 1*). These observations suggest the spatial proximity between septins and the exocyst during certain stages of cytokinesis, raising the possibility of their functional coordination, which we would further investigate below.

Since the septin and exocyst colocalize at the division site and Sec3 arrives earlier, we tested whether septin localization depends on Sec3 and other exocyst subunits. In WT cells, Spn1 always formed ring structures at the rim of division plane during septation (*Figure 1F*). In exocyst mutants *exo70Δ* and the temperature-sensitive *sec3-913* and *sec8-1*, Spn1 localization was comparable to WT at permissive temperature (*Figure 1—figure supplement 1A*). At the restrictive temperature, although Spn1 localized as a ring at the division site before septation, a fraction of Spn1 abnormally spread onto the division plane following furrow ingression in *sec3-913* and *sec8-1* mutants (*Figure 1F*, red boxes; and *Figure 1—figure supplement 1B*, middle focal plane). As *exo70Δ* cells have no severe defects (*Wang et al., 2003*), the exocyst complex may not be as compromised as in *sec3-913* and *sec8-1* mutants. Only minor mislocalization of Spn1 was observed in *exo70Δ* cells even at 36°C (*Figure 1—figure supplement 1B*). This localization pattern in exocyst mutants suggested a possible correlation between septins and the furrow ingression. Indeed, some Spn1 followed the contractile ring marked with Rng8 (*Wang et al., 2014*) as it constricted, spread onto the new plasma membrane, and concentrated at the center of the division plane while maintaining its localization at the rim (*Figure 1G*). We also examined Spn1 intensity at the division site in cells with no visible septum, forming septum, and closed septum. Spn1 levels were comparable or higher in exocyst mutants compared to WT at both 25 and 36°C (*Figure 1H*, *Figure 1—figure supplement 1C, D*). FRAP analyses of Spn1 showed

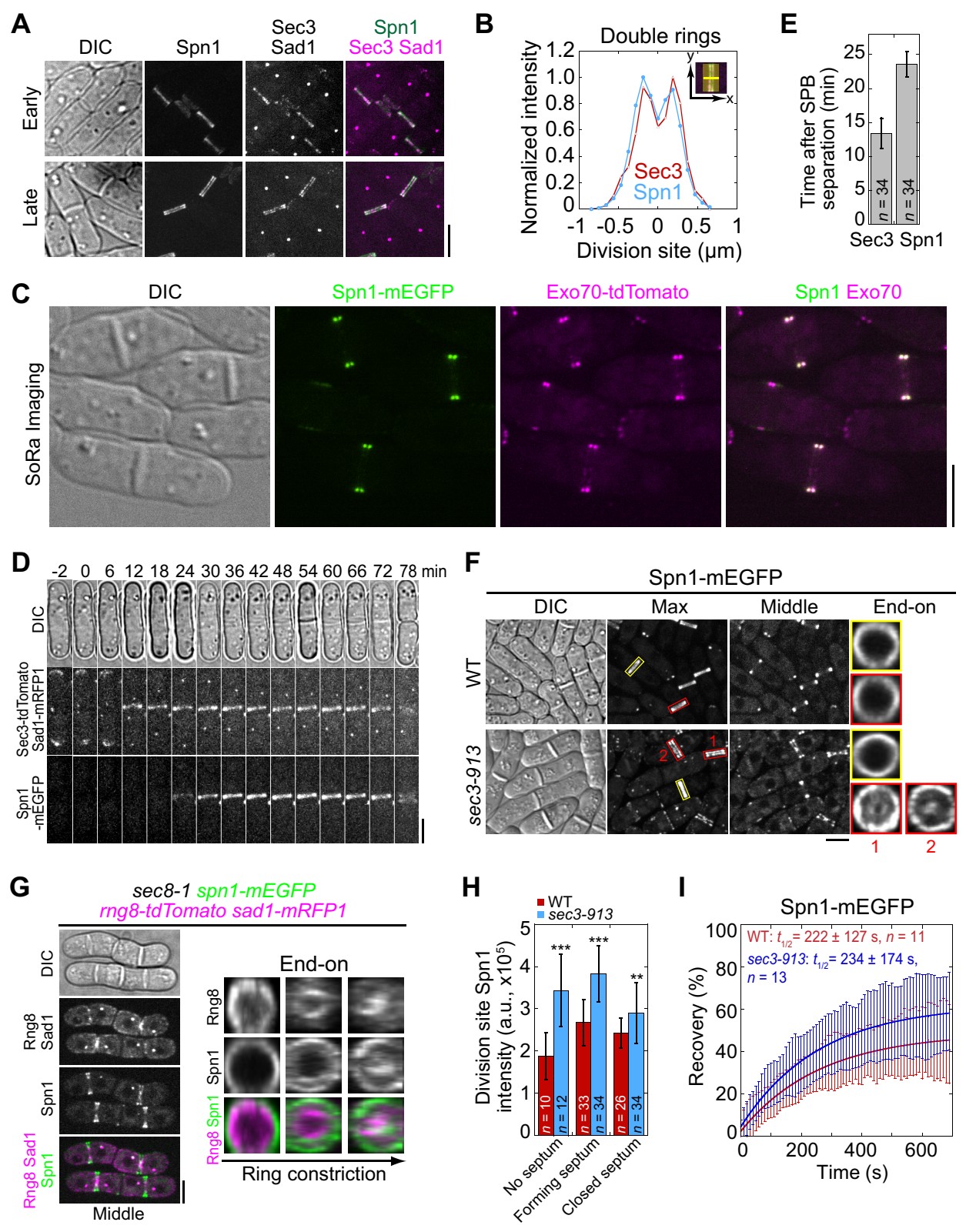

**Figure 1.** Septins and the exocyst colocalize at the division site and septins partially depend on the exocyst for localization. (**A**) Co-localization of Spn1-mEGFP and Sec3-tdTomato at the division site in cells without (early) and with (late) septa. Sad1-mRFP1 marks the spindle pole body (SPB). (**B**) Line scans showing Spn1 and Sec3 intensities at the division site along the cell long axis in septated cells as in (**A**). (**C**) SoRa (Super resolution by Optical Re-Assignment) confocal microscopy of cells expressing both Spn1-mEGFP and Exo70-tdTomato showing their perfect colocalization in the middle focal

*Figure 1 continued on next page*

*Figure 1 continued*

plane. (**D**) Time course and (**E**) quantification (in minutes) of Sec3 and Spn1 localizations and appearance timing at the division site. SPB separation is defined as time 0. (**F**) Localization of Spn1 (Max intensity projection, Middle focal plane, and End-on view of the division site) in WT and *sec3-913* cells grown at 36°C for 4 hr. Yellow boxes, cells without septa; red boxes, cells with septa. (**G**) Localization of Spn1 and the contractile-ring marker Rng8 in *sec8-1* cells grown at 36°C for 4 hr. (**H**) Spn1 intensities at the division site in WT and *sec3-913* cells grown at 36°C for 4 hr. Cells were grouped into no septum, forming septum, and closed septum stages. **p < 0.01; ***p < 0.001. (**I**) FRAP analyses (photobleached at time 0) of Spn1 at the division site in WT and *sec3-913* cells grown at 36°C for 4 hr. Mean ± SD. Bars, 5 μm.

The online version of this article includes the following video and figure supplement(s) for figure 1:

**Figure supplement 1.** Localization, intensity, and dynamics of the septin Spn1 in exocyst mutants at the division site.

**Figure 1—video 1.** Accumulation of the septin Spn1-mEGFP and the exocyst marked by Exo70-tdTomato to the division site.

https://elifesciences.org/articles/101113/figures#fig1video1

**Figure 1—video 2.** Dynamic localization of Exo70-tdTomato at the division site on a single-focal plane close to the cell surface.

https://elifesciences.org/articles/101113/figures#fig1video2

**Figure 1—video 3.** Dynamic localization of Exo70-tdTomato at the division site on the middle focal plane.

https://elifesciences.org/articles/101113/figures#fig1video3

no difference in its dynamics in WT and *sec3-913* cells at 36°C (*Figure 1I*, *Figure 1—figure supplement 1E*). Collectively, despite some Spn1 mislocalizing to the center of the division plane in exocyst mutants, the majority of Spn1 still localizes to the rim. Thus, septins only partially depend on the exocyst for their localization.

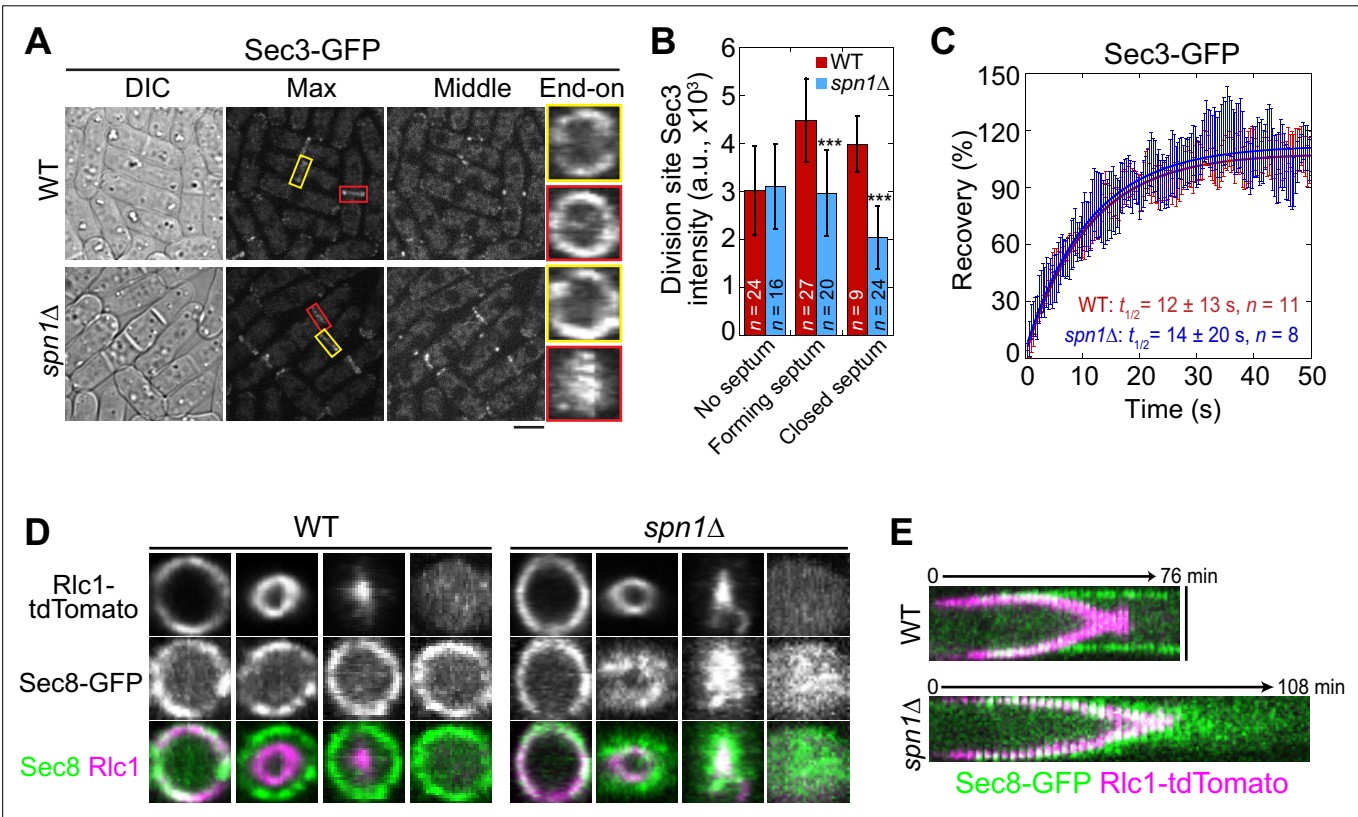

**Figure 2.** Septin rings recruit or anchor the exocyst complex to the rim of the division plane during cytokinesis. (**A**) Localization of Sec3 at the division site in WT and *spn1Δ* cells. Yellow boxes, cells without a septum; red boxes, cells with a closed septum. (**B**) Sec3 intensity at the division site in WT and *spn1Δ* cells. ***p < 0.001. (**C**) FRAP analyses of Sec3 at the division site in WT and *spn1Δ* cells. Mean ± SEM. End-on views (**D**) and kymographs (**E**) of Sec8 and the contractile ring marker Rlc1 at the division site in WT and *spn1Δ* cells. Bars, 5 μm.

The online version of this article includes the following figure supplement(s) for figure 2:

**Figure supplement 1.** Localization, intensity, and dynamics of the exocyst subunits in *spn1Δ* cells at the division site; and Sec3 and Spn1 localization in *rho4*, *gef3*, or *gef3 rho4* mutants.

Next, we examined the localization, intensity, and dynamics of the exocyst complex (subunits Sec3, Exo70, and Sec8) in *spn1Δ* cells. In mitotic cells without a septum, the exocyst localized to the rim of the division plane in both WT and *spn1Δ* cells (*Figure 2A*, *Figure 2—figure supplement 1A, B*, yellow boxes). During septation, however, the exocyst spread across the division plane as a disk in *spn1Δ* cells while it remained at the rim in WT cells (*Figure 2A*, *Figure 2—figure supplement 1A, B*, red boxes). The levels of Sec3, Exo70, and Sec8 at the division site in *spn1Δ* cells were not significantly different from WT before septation (*Figure 2B*, *Figure 2—figure supplement 1C*). During and after septum formation, the levels of all three exocyst components were significantly reduced at the division site (except Exo70 in cells with forming septum) and almost absent at the rim in *spn1Δ* cells (*Figure 2A, B*; *Figure 2—figure supplement 1A–C*). These results were confirmed in cells expressed Sec8-GFP and myosin light chain Rlc1 as a contractile ring marker (*Figure 2D, E*). However, the dynamics of Sec3 at the division site was not affected in *spn1Δ* cells (*Figure 2C*, *Figure 2—figure supplement 1D*). The exocyst was much more dynamic than septins at the division site (*Figures 1H and 2C*), which was confirmed by high temporal resolution imaging of Exo70 (*Figure 1—videos 2 and 3*). Together, loss of septins results in exocyst mislocalization and its decreased levels, especially at the rim of the division plane during septum formation, suggesting that septins play an important role in regulating the exocyst localization at the division site.

Collectively, our data suggest that septins and the exocyst complex colocalize and are interdependent for localization at the division site during and after the contractile ring constriction, with the septin rings being more important for the exocyst localization. Thus, we conclude that the initial recruitment of the exocyst to the division site does not depend on septins, but its rim localization and maintenance during late cytokinesis require the septin rings.

## Septins regulate the exocyst localization through physical interactions

Septins have been shown to play a role in the Rho GEF Gef3–Rho4 GTPase pathway to regulate the exocytosis of glucanases Eng1 and Agn1 for proper cell separation (*Pérez et al., 2015*; *Wang et al., 2015*). Septins are essential for Gef3 localization to the division site (*Muñoz et al., 2014*; *Wang et al., 2015*). In *spn1Δ* cells, Gef3 localization on the plasma membrane is abolished, and Rho4 localization in cells with a closed septum was significantly reduced (*Wang et al., 2015*). Since Rho4 can interact with both the exocyst complex and septins (*Pérez et al., 2015*), we tested whether the altered exocyst localization pattern that we observed in septin mutants was through Gef3 and Rho4. We found that Spn1 ring localization was not affected in *rho4Δ gef3Δ* cells (*Figure 2—figure supplement 1E*). Although partially mislocalized to the center of the division plane, the majority of Sec3 still localized as a ring at the rim of the division plane in *rho4Δ*, *gef3Δ*, and *rho4Δ gef3Δ* cells (*Figure 2—figure supplement 1F*). Thus, the different localization patterns of the exocyst in *spn1Δ* and *rho4Δ gef3Δ* cells suggest that septins can regulate exocyst localization independent of Gef3 and Rho4.

To test the hypothesis that septins regulate the localization of the exocyst directly, we examined the physical interactions between septins and the exocyst subunits. Sec3 and Exo70 are the most important subunits for the targeting of the octameric exocyst to the plasma membrane (*Boyd et al., 2004*; *He et al., 2007*; *Bendezú et al., 2012*; *Luo et al., 2014*; *Yue et al., 2017*; *Liu et al., 2018*; *Synek et al., 2021*), and Spn1 and Spn4 are essential for septin localization and functions (*An et al., 2004*). Therefore, we first tested the interactions between Spn1–Sec3, Spn1–Exo70, Spn4–Sec3, and Spn4–Exo70 using co-immunoprecipitation of cell extracts from fission yeast. Surprisingly, no physical interactions were detected among the four proteins.

Then we utilized AlphaFold2_advanced ColabFold algorithm (*Jumper et al., 2021*; *Mirdita et al., 2022*), whose highly accurate predictions of protein structures have revolutionized structural biology, to predict the physical interactions between all 32 combinations of the four septins and eight exocyst subunits. For the modeling, the complete sequences of each subunit of septins and exocyst complex were used except Sec8. Sec8 subunit was analyzed in two fragments with overlapping sequence due to the 1400 amino acids input sequence limitation of AlphaFold2_advanced. The generated models with the highest confidence are shown (*Figure 3*, *Figure 3—figure supplement 1*). Predicted interacting interface residues that were defined as amino acids of two possible binding partners with distance ≤4 Å were calculated from the rank 1 predicted model (*Yin and Pierce, 2024*). Then the contact residues were further narrowed down by excluding the residues having predicted local-distance difference test (pLDDT) score <50. Based on the above analyses, we predicted the following

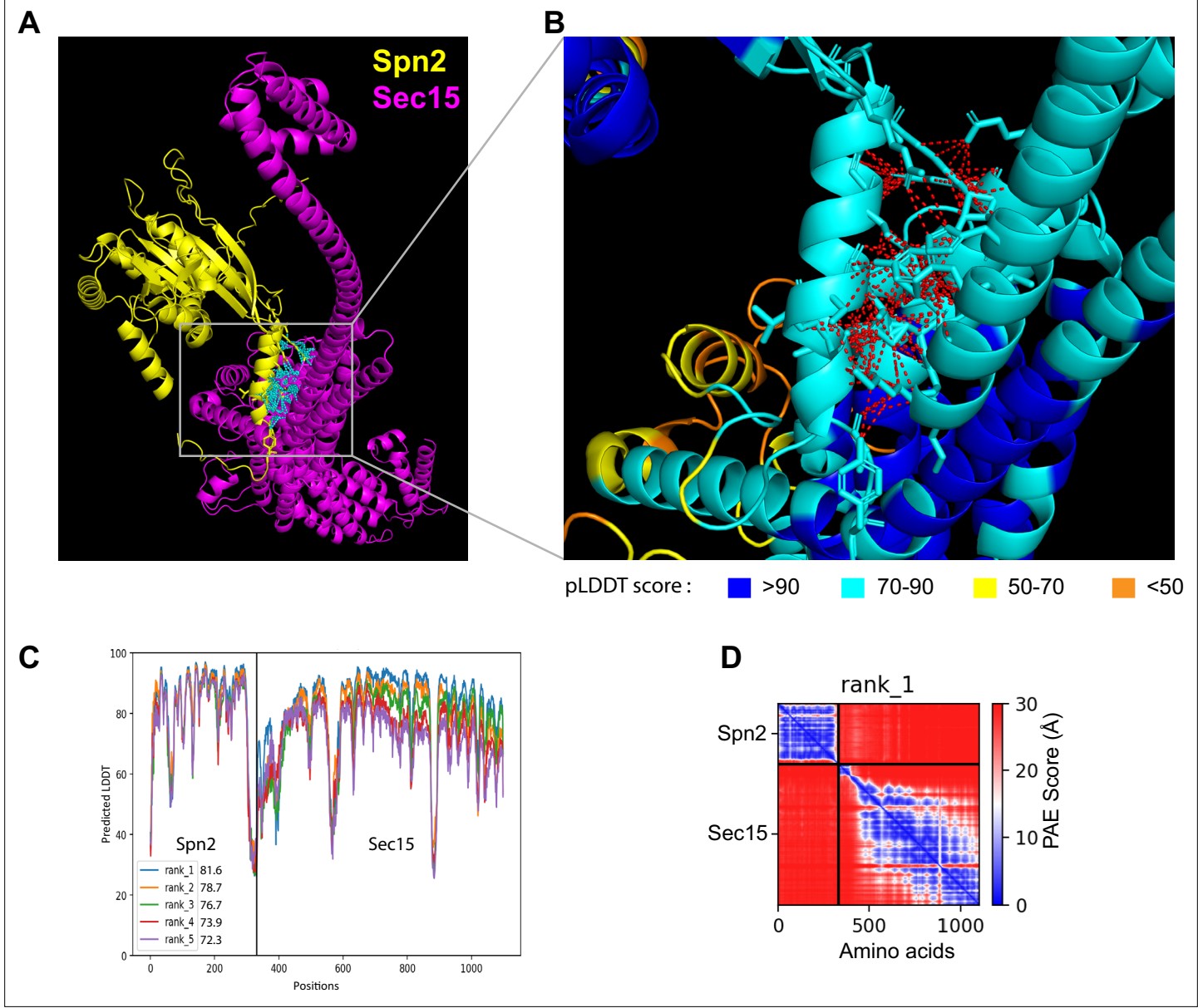

**Figure 3.** The 3D structural model of predicted interactions between Spn2 and Sec15 generated by AlphaFold. (**A, B**) AlphaFold2_advanced predicted interaction between Spn2 and Sec15 in rank 1 model with predicted local-distance difference test (pLDDT) score of 81.6. The pTM value = 0.51. Spn2 is colored in yellow and Sec15 in magenta. (**B**) Inset of enlarged view of the predicted interactions, contacts between interface residues with distance <4 Å are colored in red (those in cyan in A). Residues are colored corresponding to their pLDDT scores as indicated in the legends below. (**C**) Residue position scores of five predicted models for Spn2 and Sec15 interactions ranked according to pLDDT scores. (**D**) PAE (Predicted Alignment Error) plot for the top-ranked model shown in (**A–C**), where colors represent confidence in the relative positioning of residues across the two proteins. Lower values (blue) represent high confidence while higher values (red) show low confidence in domain–domain interactions.

The online version of this article includes the following video and figure supplement(s) for figure 3:

**Figure supplement 1.** The 3D structural models of septin–exocyst interactions generated by AlphaFold.

**Figure supplement 2.** The predicted 3D structural model of *S. pombe* exocyst complex by AlphaFold3, highlighting the residues that interact with septins (also see *Figure 3—video 1*).

**Figure supplement 3.** The predicted 3D structural models of *S. pombe* septin complexes by AlphaFold3, highlighting the exocyst-interacting residues (also see *Figure 3—video 2*; *Figure 3—video 3*; *Figure 3—video 4*).

**Figure 3—video 1.** The predicted 3D structural model of *S. pombe* exocyst complex by AlphaFold3, highlighting the residues that interact with septins.
https://elifesciences.org/articles/101113/figures#fig3video1

*Figure 3 continued on next page*

*Figure 3 continued*

**Figure 3—video 2.** The predicted 3D structural model of *S. pombe* septin octameric complex by AlphaFold3, highlighting the exocyst-interacting residues.

https://elifesciences.org/articles/101113/figures#fig3video2

**Figure 3—video 3.** The predicted 3D structural model of *S. pombe* septins Spn1, Spn2, and Spn4 hexameric complex by AlphaFold3, highlighting the exocyst-interacting residues.

https://elifesciences.org/articles/101113/figures#fig3video3

**Figure 3—video 4.** The predicted 3D structural model of *S. pombe* septins Spn1, Spn3, and Spn4 hexameric complex by AlphaFold3, highlighting the exocyst-interacting residues.

https://elifesciences.org/articles/101113/figures#fig3video4

top six interactions between the septin and exocyst subunits: Spn2 and Sec15 (*Figure 3*), Sec15 and Spn1, Sec6 and Spn1, Spn2 and Sec5, Spn4 and Sec15, and Spn4 and Sec3 (*Figure 3—figure supplement 1*).

To test whether the interacting residues calculated from the pair-wise predictions are accessible in the whole exocyst or septin complexes, we employed AlphaFold3 and the cryo-EM structure of the whole *S. cerevisiae* exocyst complex (with 4.4 Å resolution) to examine the interfaces of septins and the exocyst interactions, assuming that the *S. pombe* exocyst has a similar structure (*Mei et al., 2018*). For the septin complex, we first predicted its octameric structure using two copies of each fission yeast Spn1–4. We examined all the interacting residues on the septin and exocyst complex predicted from our AlphaFold2 modeling to determine whether these predicted interactions are structurally compatible (*Figure 3—figure supplements 2 and 3A*; *Figure 3—video 1*; *Figure 3—video 2*). Our analyses revealed that 84% of exocyst and 96% of septin predicted interacting residues were sterically feasible without disrupting the architecture of exocyst or septin complex (the residues highlighted in yellow in *Figure 3—figure supplement 2*, *Figure 3—figure supplement 3*; *Figure 3—video 1*; *Figure 3—video 2*), while others would likely require partial disassembly or flexible conformations. Because septins can also form hexameric complexes (*McMurray and Thorner, 2019*; *Mendonça et al., 2019*; *Soroor et al., 2021*), we also predicted hexameric septin complexes without either Spn2 or Spn3 and found that 86% or 92% exocyst-interacting residues are available on the surface, respectively (*Figure 3—figure supplement 3B, C*; *Figure 3—video 3*; *Figure 3—video 4*). We did not predict a septin hexameric complex without Spn1 or Spn4, as they are the core subunits responsible for septin localization and functions at the division site (*An et al., 2004*; *Onishi et al., 2010*; *Wu et al., 2010*). These predictions indicate that these septin–exocyst interactions are sterically plausible.

Next, we used reciprocal Co-IP assays of fission yeast extracts to confirm the predicted interactions between septin and exocyst subunits. Out of the six predicted interactions, we found five of them were positive in Co-IP. We found that Spn2 physically interacted with Sec15 and Sec5, Spn1 with Sec15 and Sec6, and Spn4 with Sec15 (*Figure 4*, *Figure 4—figure supplement 1*). Sec15 interacted with three septins Spn1, Spn2, and Spn4, which were stronger than other combinations. We also utilized yeast two-hybrid assays to confirm these five pairs of interactions (*Figure 4E, F*). X-gal overlay assay (insets) and quantification of β-galactosidase using o-nitrophenyl-β-D-galactopyranoside (ONPG) suggested that Sec15 may directly interact with Spn1, Spn2, and Spn4 (*Figure 4E*); and Sec6 interacted with Spn1 through its C-terminal fragment (Spn1[300–469]) that contains the coiled-coil motif (*Figure 4F*). The Spn2–Sec5 interaction could not be tested due to a very high level of autoactivation of Sec5. Thus, we conclude that septins physically and likely directly interact with the exocyst in fission yeast via multivalent interactions.

## Septins are involved in concentrating Sec15 and Sec5 at the rim of the division plane, especially during the late stage of cytokinesis

We reasoned that septins localize the exocyst at the division site via their multivalent interactions with the exocyst subunits Sec15, Sec5, and Sec6. Weakened interactions between septins and the exocyst in the absence of a certain septin subunit could lead to mislocalization of the exocyst complex. Indeed, similar to the results presented in *Figures 1 and 2*, *Figure 1—figure supplement 1*, and *Figure 2—figure supplement 1* with other exocyst subunits, the deletion of *spn1* or *spn4* led to mislocalization of Sec15 on the division plane in ~75% of cells with a septum while Sec15 in ~90% of WT cells

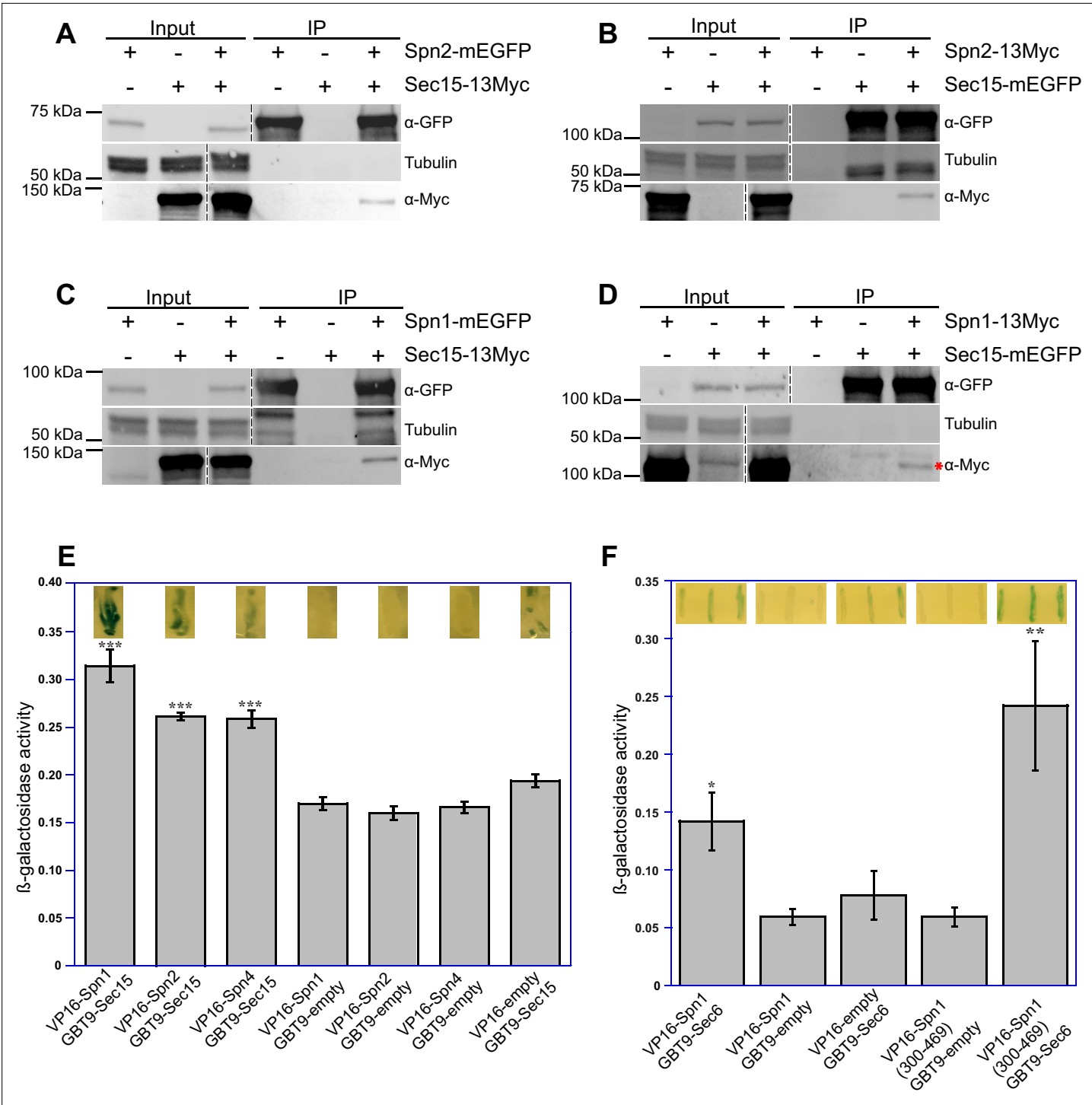

**Figure 4.** Septins and the exocyst interact physically. Reciprocal co-immunoprecipitation of Sec15 with Spn2 (**A, B**) and Spn1 (**C, D**). Septin or exocyst subunits tagged with mEGFP or 13Myc were immunoprecipitated using antibodies against GFP from cell lysates, separated on SDS–PAGE, and incubated with appropriate antibodies. Tubulin was used as a loading control. Asterisk (*) in (**D**) marks Spn1-13Myc. The vertical dashed lines mark the positions of protein ladders that were excised out. $n = 3$. (**E, F**) Septins and the exocyst subunits may interact directly, revealed by the yeast two-hybrid assays. X-gal overlay results (insets on the top of the columns) and quantification of β-galactosidase activities using o-nitrophenyl-β-D-galactopyranoside (ONPG) showing interactions between (**E**) Sec15 with Spn1, Spn2, and Spn4; and (**F**) Sec6 with Spn1 and its coiled–coil motif Spn1(300–469). Data is shown in Mean ± SD, $n = 3$ (in E) or 4 (in F). ***$p ≤ 0.0001$, **$p ≤ 0.001$, *$p ≤ 0.01$ compared with their respective controls in one-way ANOVA with Tukey's post hoc test.

The online version of this article includes the following source data and figure supplement(s) for figure 4:

*Figure 4 continued on next page*

*Figure 4 continued*

**Source data 1.** Raw western blot images unlabeled.

**Source data 2.** Raw western blot images labeled.

**Figure supplement 1.** Septins and the exocyst interact physically.

**Figure supplement 1—source data 1.** Raw western blot images unlabeled.

**Figure supplement 1—source data 2.** Raw western blot images labeled.

localized as rings at the rim of the division plane in septating cells (*Figure 5A, B*). Results from time-lapse microscopy of *spn1Δ* or *spn4Δ* cells were consistent with these findings. Sec15 was first recruited to the division site as rings and then spread to the whole division plane before signal disappearance, leading to some multiseptated cells (*Figure 5—video 1*; *Figure 5—video 2*; *Figure 5—video 3*).

Although septin filaments have four subunits in vegetative cells (*An et al., 2004*), Spn2 is less important than Spn1 and Spn4 for septin functions, and *spn2Δ* has a much weaker phenotype in septation than *spn1Δ* or *spn4Δ* (*An et al., 2004*; *Wu et al., 2010*; *Zheng et al., 2018*). Consistently, in *spn2Δ* cells, Sec15 and Sec5 localized normally at the division site before septation (*Figure 5A, C*). Both Sec15 and Sec5 spread more or less to the whole division plane in ~50% of *spn2Δ* cells with obvious septa (*Figure 5A–D*). Unlike in *spn1Δ* or *spn4Δ* cells, a fraction of Sec15 and Sec5 still localized to the rim in *spn2Δ* cells with a septum (*Figure 5A, C*). Collectively, these data support the conclusion that Spn1, Spn2, and Spn4 are important for targeting the exocyst to the rim of division plane during cytokinesis through physical interactions.

## Septin mutants affect the sites of secretory vesicle tethering and cargo delivery at the division plane

Septin and exocyst mutations showed no or very mild synthetic genetic interactions (*Tables 1 and 2*), suggesting that septins and the exocyst complex function in the same pathway to regulate cytokinesis and septation. Surprisingly, they had different genetic interactions with the transport particle protein-II (TRAPP-II) mutants (*Tables 1 and 2*). The exocyst mutant *sec8-1* is synthetic lethal with *trs120-M1* and has severe synthetic cytokinesis defects with *trs120-ts1* due to the overlapping function of the exocyst and TRAPP-II in exocytosis during fission yeast cytokinesis (*Wang et al., 2016*). However, *spn1Δ trs120-M1* and *spn1Δ trs120-ts1* double mutants were viable with no obvious synthetic interactions (*Tables 1 and 2*). Thus, septins and the exocyst also work in different genetic pathways for certain functions in fission yeast.

The exocyst complex is the major tether of secretory vesicles at the plasma membrane (*TerBush and Novick, 1995*; *TerBush et al., 1996*; *Wang et al., 2002*; *Luo et al., 2014*). So, we tested whether exocyst mislocalization in septin mutants compromises the targeting of secretory vesicles and their cargos. We first performed electron microscopy to examine if secretory vesicles are accumulated at the division site in *spn1Δ* cells (*Figure 6A*). During septum formation, seven- and twofold more secretory vesicles accumulated at the division site in *sec8-1* and *spn1Δ* cells, respectively, compared to WT (*Figure 6A, B*). However, in cells with a closed septum, the number of secretory vesicles adjacent to the division site was not significantly different between WT and *spn1Δ* cells (*Figure 6B*). Consistently, secretory vesicle markers Rab11 GTPase Ypt3 and vSNARE Syb1 accumulated more in the center of the division plane but diminished from the rim in *spn1Δ* cells (*Figure 6C, D*). The accumulation of the secretory vesicles at the division plane and their mistargeting are consistent with exocyst mislocalization in *spn1Δ* cells.

We next examined the distribution of two secretory vesicle cargos, the β-glucan synthase Bgs1/Cps1 and the β-glucanase Eng1, which are delivered to the division site by secretory vesicles during cytokinesis (*Liu et al., 1999*; *Baladrón et al., 2002*; *Cortés et al., 2002*; *Martín-Cuadrado et al., 2003*). More Bgs1 localized in the center of the division plane in *spn1Δ* cells compared to WT (*Figure 7A*). *spn1Δ* and *sec8-1* cells also had thicker septa compared to WT cells (*Figure 7B*). Another cargo of secretory vesicles, Eng1, spread across the division plane as a disk with localization clearly missing at the rim in *spn1Δ* cells (*Figure 7C*). Lack of the glucanase Eng1 at the rim could contribute to the delayed cell separation in *spn1Δ* cells since the junctions between septum and the cell wall cannot be efficiently digested, consistent with earlier studies (*Baladrón et al., 2002*; *Martín-Cuadrado et al.,*

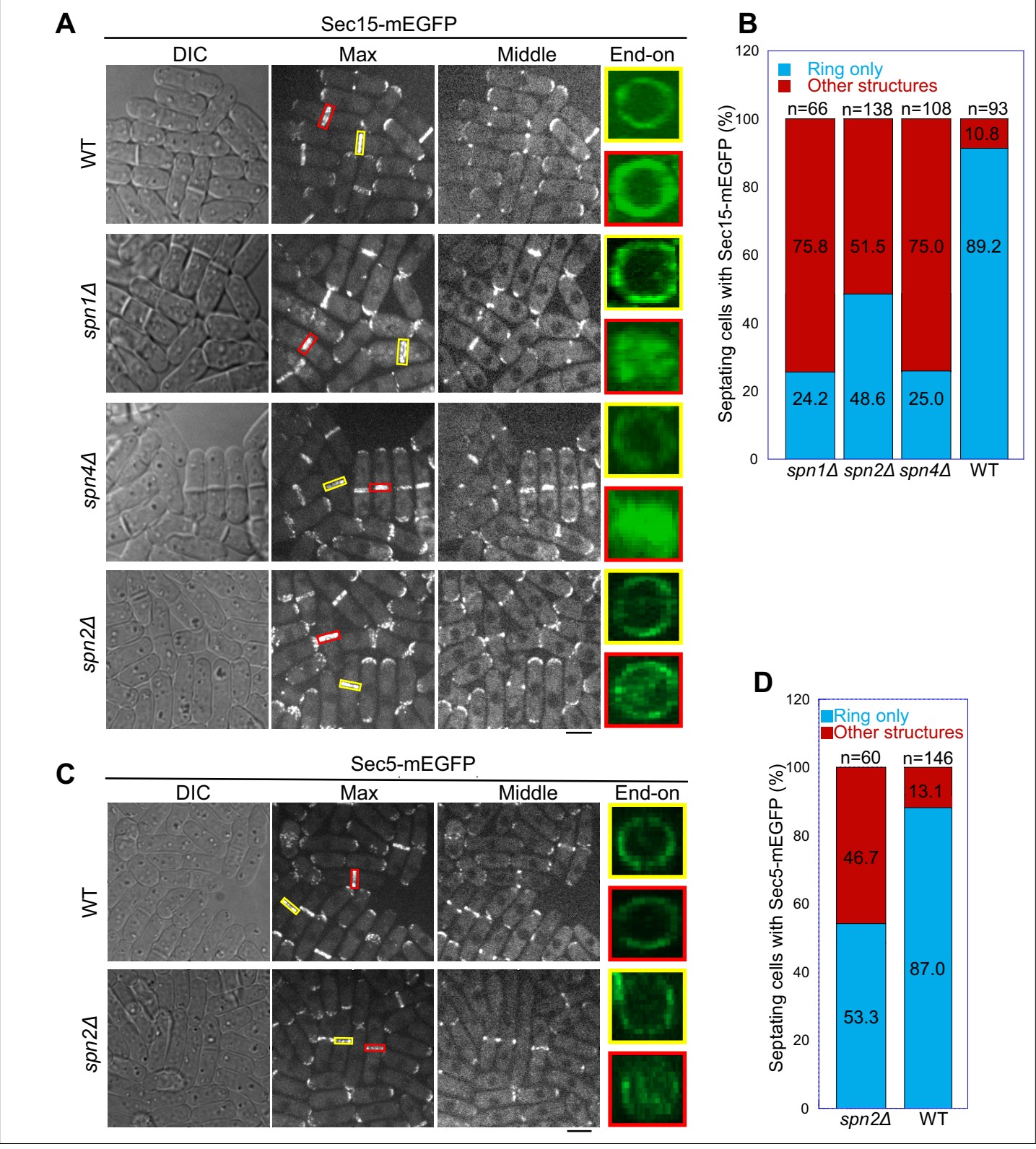

**Figure 5.** Localization patterns of both Sec15 and Sec5 at the division site depend on septins. Localization of (**A, B**) Sec15 and (**C, D**) Sec5 at the division site in WT and septin mutant cells. Yellow boxes, cells without a septum; red boxes, cells with a closed septum in (**A, C**). Quantification of cells with intact and mislocalized Sec15 (**B**) and Sec5 (**D**) signals in WT and septin mutant cells with obvious septa. Scale bars, 5 μm.

The online version of this article includes the following video(s) for figure 5:

*Figure 5 continued on next page*

*Figure 5 continued*

**Figure 5—video 1.** Localization of Sec15 as a ring at the rim of the division plane during cytokinesis in WT cells.
https://elifesciences.org/articles/101113/figures#fig5video1

**Figure 5—video 2.** Mislocalization of Sec15 as a disk at the division site in *spn1Δ* cells.
https://elifesciences.org/articles/101113/figures#fig5video2

**Figure 5—video 3.** Mislocalization of Sec15 as a disk at the division site in *spn4Δ* cells.
https://elifesciences.org/articles/101113/figures#fig5video3

*2003*). Our studies on Bgs1 and Eng1 indicate an increase of vesicle tethering in the center and a loss at the rim of the division plane without septins.

Collectively, our data indicate that septins play important roles in maintaining the proper localization and targeting of the exocyst on the plasma membrane during cytokinesis. Loss of septins results in spreading of the exocyst across the division plane and tethering of secretory vesicles at the wrong destination, which leads to the accumulation of secretory vesicles and mistargeting of downstream cargos.

## Discussion

In this study, we reveal that septins and the exocyst complex physically interact to regulate exocytosis and ensure proper targeting of vesicle cargos to the plasma membrane during cytokinesis.

### Septins are important for proper membrane targeting of the exocyst complex to ensure successful cytokinesis

Septins are essential for cytokinesis and other cellular processes in budding yeast and many other organisms (*Neufeld and Rubin, 1994*; *Longtine et al., 1996*; *Kinoshita et al., 1997*; *Gladfelter et al., 2001*; *Russell and Hall, 2005*; *Oh and Bi, 2011*). However, the nature of their functions is only partially understood. It has been a mystery why the phenotypes of septin mutants are quite mild in fission yeast ever since their discoveries in the early 1990s, yet their sequences and structures are evolutionarily conserved across species (*Longtine et al., 1996*; *An et al., 2004*; *Zheng et al., 2018*; *Zheng et al., 2024*). In this study, we investigated the spatial regulation of the exocyst complex and roles of septins during cytokinesis in the fission yeast model system. Without septin rings, the exocyst complex, which specifies for the sites for vesicle fusion on the plasma membrane, cannot

**Table 1.** Viability of double mutants of the septin and exocyst from tetrad dissection at 25°C.

| Parent 1 | Parent 2 | Viable double mutants (%) at 25°C* | Total number of tetrads |
|---|---|---|---|
| *spn1Δ* | *sec3-916* | 78 | 14 |
| *spn1Δ* | *sec3-913* | 100 | 13 |
| *spn1Δ* | *sec8-1* | 100 | 12 |
| *spn1Δ* | *exo70Δ* | 100 | 14 |
| *spn1Δ* | *trs120-M1* | 100 | 27 |
| *spn1Δ* | *trs120-ts1* | 95 | 18 |
| *spn2Δ* | *sec3-916* | 83 | 10 |
| *spn2Δ* | *sec3-913* | 100 | 10 |
| *spn3Δ* | *sec3-916* | 100 | 12 |
| *spn3Δ* | *sec3-913* | 75 | 11 |
| *spn4Δ* | *sec3-916* | 80 | 10 |
| *spn4Δ* | *sec3-913* | 100 | 14 |
| *spn4Δ* | *sec8-1* | 100 | 11 |

*Percentage of viable double mutant colonies after tetrad dissection and grown at 25°C.

**Table 2.** Genetic interactions between septin and exocyst mutations at various temperatures[*].

| Mutations | 25°C | 30°C | 32°C | 36°C |
|---|---|---|---|---|
| sec3-916 | +++[†] | ++[‡] | +[§] | -[¶] |
| sec3-913 | +++ | +++ | ++ | - |
| sec8-1 | +++ | ++ | + | - |
| spn1Δ | ++ | ++ | ++ | ++ |
| spn1Δ sec3-916 | ++ | + | - | - |
| spn1Δ sec3-913 | ++ | ++ | ++ | - |
| spn1Δ sec8-1 | ++ | ++ | + | - |
| exo70Δ | +++ | +++ | ++ | - |
| spn1Δ exo70Δ | ++ | ++ | ++ | - |
| trs120-M1 | ++ | - | - | - |
| spn1Δ trs120-M1 | ++ | - | - | - |
| trs120-ts1 | +++ | +++ | + | - |
| spn1Δ trs120-ts1 | ++ | ++ | + | - |
| spn2Δ | +++ | +++ | +++ | +++ |
| spn2Δ sec3-916 | ++ | + | + | - |
| spn2Δ sec3-913 | +++ | ++ | + | - |
| spn3Δ | +++ | +++ | +++ | +++ |
| spn3Δ sec3-916 | ++ | ++ | + | - |
| spn3Δ sec3-913 | +++ | ++ | ++ | - |
| spn4Δ | ++ | ++ | ++ | ++ |
| spn4Δ sec3-916 | ++ | + | - | - |
| spn4Δ sec3-913 | ++ | ++ | ++ | - |
| spn4Δ sec8-1 | ++ | ++ | ++ | - |

[*]Cells were freshly grown on YE5S and YE5S + Phloxin B (which accumulates in dead cells) plates before checking the growth and morphology under DIC at different temperatures. The defects in cytokinesis and cell integrity compared with the parent strains were classified as follows:

[†]+++, comparable to wt.

[‡]++, some cell lysis or cytokinesis defects.

[§]+, severe cytokinesis defects with reduced growth.

[¶]-, inviable.

maintain its localization at the rim of the division plane. Instead, the exocyst complex follows actomyosin contractile ring constriction and spreads across the whole division plane. Although loss of septins does not affect the dynamics of the exocyst, the targeting sites of secretory vesicles and their cargos are altered, which may contribute to a thicker septum and a delayed cell separation. The modest accumulation of vesicles and vesicle cargos at the division site is one of the reasons for the increased thickness of the division septum in septin mutants. It is more likely that the misplaced exocyst can still tether vesicles along the division plane without septins. Due to the lack of the glucanase Eng1 at the rim of the division plane in septin mutants, daughter-cell separation is delayed, and then cells continue to thicken the septum. The relatively modest vesicle accumulation in septin mutants compared to the exocyst mutant suggests that septins are not absolutely required for vesicle tethering or fusion per se at the division site. Instead, septins primarily function to spatially organize the targeting sites of exocyst-directed vesicles by stabilizing the localization of the exocyst at the rim of the cleavage furrow. In septin mutants, mislocalization of the exocyst reduces the spatial precision of membrane insertion but still permits vesicle tethering and fusion, albeit in a less controlled manner. Thus, septins likely play a modulatory rather than essential role in exocytic vesicle delivery during cytokinesis. This

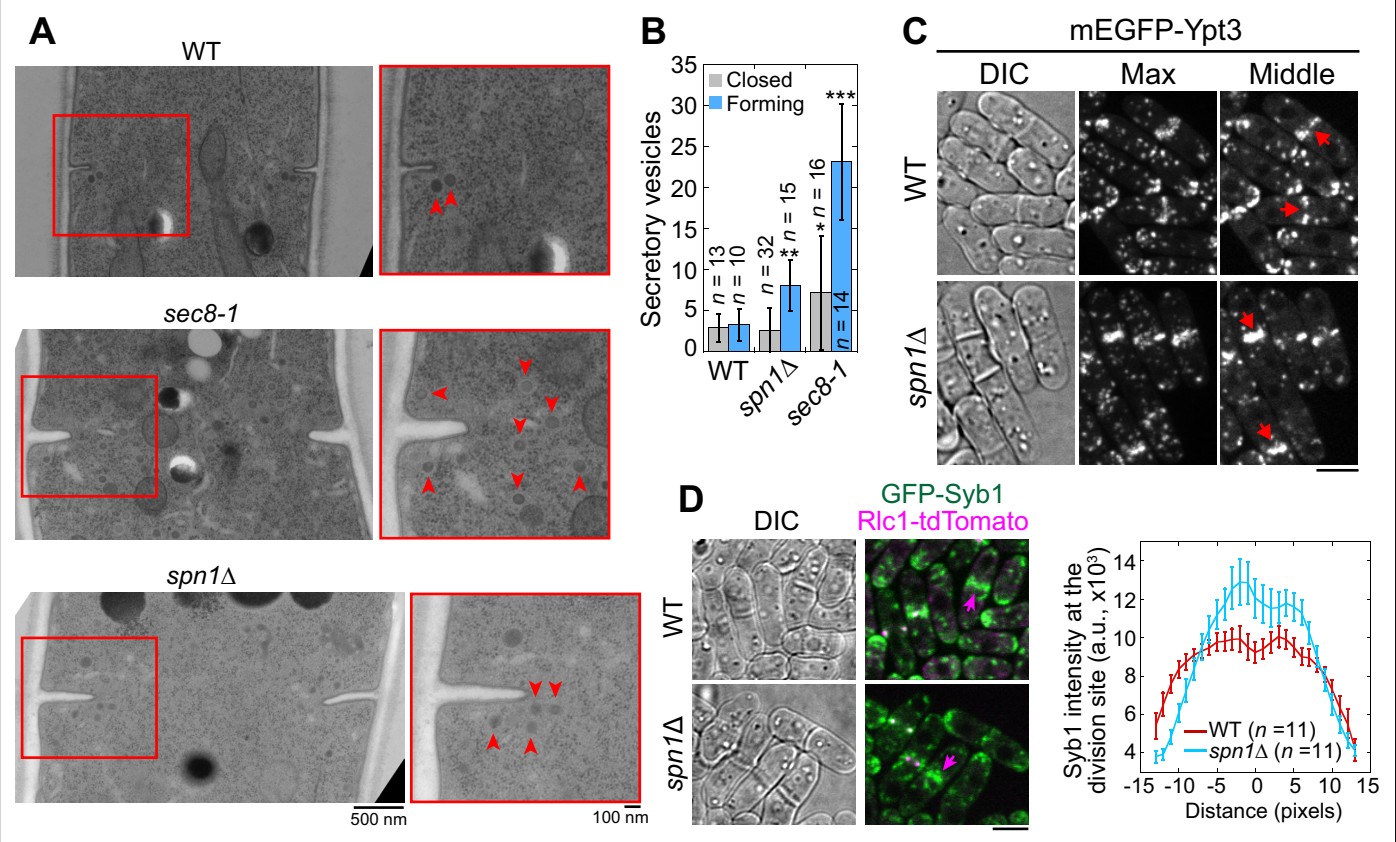

**Figure 6.** Septins are important for proper localization and distribution of secretory vesicles. EM thin-section images (**A**) and quantifications of secretory vesicles (**B**) in WT, *sec8-1*, and *spn1Δ* cells with forming or closed septa. Cells were grown at 36°C for 4 hr. Red boxes indicate the enlarged regions on the right. Arrowheads mark secretory vesicles. *p < 0.05; **p < 0.001; ***p < 0.0001 compared to WT. n = numbers of thin sections. Localizations of the Rab11 GTPase Ypt3 (**C**) and the v-SNARE Syb1 and Rlc1 (**D**) in WT and *spn1Δ* cells. Arrows mark examples of cells with closed septa. Syb1 intensities at the division site (D, right) from line scans at the middle focal plane of cells with closed septa (at the end of ring constriction indicated by an Rlc1 dot at the center of the division plane). Bars, 500 nm (**A**, left), 100 nm (**A**, right), and 5 μm (**C, D**).

interpretation aligns with our localization and genetic interaction data, which indicates that septins act as scaffolds to optimize secretion geometry, rather than as core components of the fusion machinery. Thus, fission yeast septins function in exocytosis through maintaining proper docking sites of the exocyst complex and secretory vesicles at the division site.

Both the exocyst and TRAPP-II complex tether vesicles at the cleavage furrow during cytokinesis (*Wang et al., 2016*). The genetic interactions between mutations in the exocyst and septins when combined with TRAPP-II mutants may reflect fundamentally different consequences for compromising the exocyst function (*Tables 1 and 2*). In septin mutants, the exocyst complex still localizes to the division site but is mispositioned from the rim to the center of the division plane. This mislocalization allows partial retention of exocyst function, leading to very mild synthetic or additive defects when combined with compromised TRAPP-II trafficking and tethering. In contrast, in exocyst subunit mutants, the exocyst becomes partial or non-functional, resulting in a more severe loss of exocyst activity. These differing consequences could explain the qualitative differences in genetic interactions observed with TRAPP-II mutants (*Tables 1 and 2*). Thus, septins and the exocyst also work in different genetic pathways for certain functions in fission yeast cytokinesis.

Fission yeast septins regulate the exocyst in specific temporal and spatial manners. They only regulate the localization of the exocyst during contractile-ring constriction and septum formation and are not responsible for its targeting to the cell tips during interphase or initial recruitment to the division site during early cytokinesis before septin appearance (*Figures 1D and 2A*; *Figure 2—figure supplement 1A, B*; and *Figure 1—video 1*; *Figure 1—video 2*; *Figure 1—video 3*). Disruption of the contractile ring affects the localization of the exocyst to the division site (*Wang et al., 2002*;

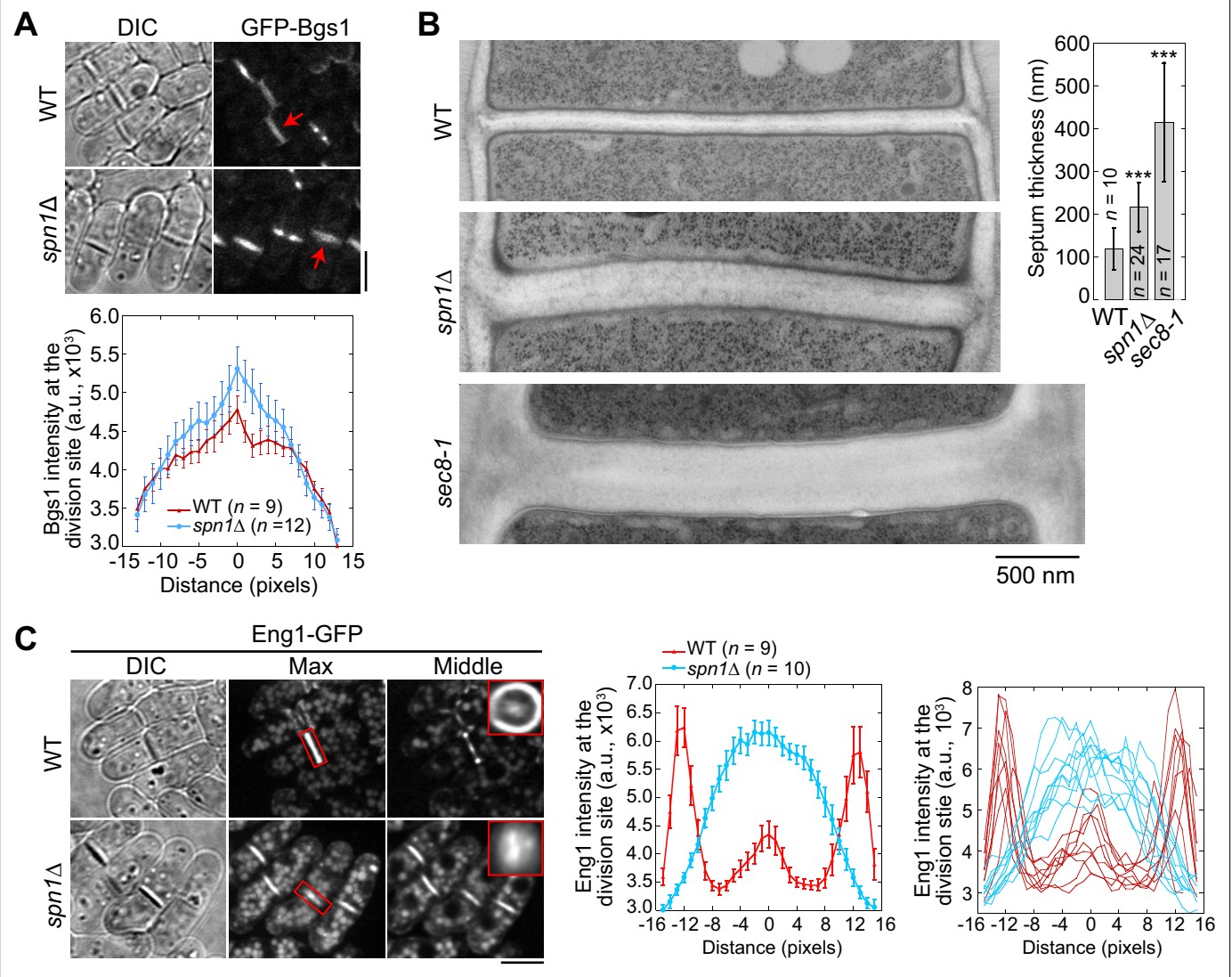

**Figure 7.** Septins are important for localization and distribution of secretory cargos Bgs1 and Eng1. (**A**) Localization (top) and intensity (bottom) of the glucan synthase Bgs1 in WT and *spn1Δ* cells. Arrows mark examples of cells with a closed septum. Bgs1 intensities from line scans across the division site at the middle focal plane were compared in cells with closed septa. (**B**) EM thin-section images (left) and septum thickness (right) of WT, *spn1Δ*, and *sec8-1* cells with closed septa. Cells were grown at 36°C for 4 hr. ***p < 0.0001 compared to WT. (**C**) Localization (left) and intensity (middle and right) of Eng1-GFP in WT and *spn1Δ* cells. The end-on views of Eng1 at the division site in cells with closed septa are shown as insets. Eng1 intensities (middle, mean intensities; and right, individual cells) are from line scans at the middle focal plane. Bars, 5 μm (**A, C**) and 500 nm (**B**).

*Dobbelaere and Barral, 2004*). This suggests that the exocyst likely depends on the contractile ring components for initial recruitment to the division site. However, this is not a universal mechanism. The subcellular localization of the exocyst complex in rat brain cells is affected by microtubule, but not actin-disrupting drugs (*Vega and Hsu, 2003*). Thus, how the exocyst is initially recruited to the division site remains to be studied. Since fission yeast exocyst clearly depends on septins for proper localization during later stages of cytokinesis, its localization dependence must migrate to septin rings from the contractile ring at some point before the onset of the contractile ring constriction. So it will be of great interest to examine how this transition occurs. Although septins may act as either scaffolds or diffusion barriers for Sec3 in budding yeast, Sec3 localizes to the region between the split septin rings during cytokinesis (*Dobbelaere and Barral, 2004*). However, in mammalian neurons, the exocyst subunits Sec6 and Sec8 colocalize with the septin SEPT7/CDC10 (*Hsu et al., 1998*). Thus, the colocalized septins and the exocyst in fission yeast may provide more insights in mammalian cells for understanding the molecular mechanisms of their interactions.

Examples of localization dependence between septins and the exocyst have been reported in other systems. The most prominent cases come from fungal pathogens. *Magnaporthe oryzae* infects plants through a specialized infection cell called appressorium, which breaches through the cuticle of the leaf to allow entry into plant tissues (*Dagdas et al., 2012*; *Gupta et al., 2015*; *Zhang et al., 2021*). The exocyst assembles in the appressorium at the point of plant infection in a septin-dependent manner. Septin deletion causes mislocalization of the key component for the exocyst assembly, Sec6, at the appressorium pore (*Gupta et al., 2015*). Similarly, the root-infecting phytopathogenic fungus *Verticillium dahliae* also assembles the exocyst at the penetration peg of the hyphopodium in a septin-dependent manner (*Zhou et al., 2017*). The absence of septin VdSep5 impairs the delivery of secretory proteins to the penetration interface (*Zhou et al., 2017*). Another example is *Candida albicans* septins, which localize at the hyphal tips where tip growth occurs with active exocytosis in this human opportunistic pathogen (*Li et al., 2007*). Deletion of septin *CDC10* or *CDC11* causes mislocalization of the exocyst marked by Sec3 (*Li et al., 2007*). Thus, one of the conserved roles of septins is to regulate the proper membrane targeting of the exocyst complex to the plasma membrane and to ensure spatiotemporal fidelity of vesicle tethering and fusion. Our current study will provide insights into how septins and the exocyst help fungal pathogens infect their hosts. However, how they physically interact with each other had not been systematically investigated in fungal pathogens.

## The exocyst complex docks on septins on the plasma membrane through multivalent physical interactions

Despite the relationships between septins and the exocyst mentioned above, whether and how they physically interact with each other was obscure. In budding yeast, the exocyst subunits have been shown to interact physically with a number of proteins, including Sec15 with Rab GTPase Sec4 and type V myosin Myo2; and Sec6 with v-SNARE protein Snc2, t-SNARE protein Sec9, and Sec1/Munc18 family protein Sec1 (*Guo et al., 1999*; *Sivaram et al., 2005*; *Jin et al., 2011*; *Shen et al., 2013*; *Lepore et al., 2016*). The septin dynamics are essential for exocytosis (*Tokhtaeva et al., 2015*). But septins and the exocyst do not colocalize in budding yeast (*Dobbelaere and Barral, 2004*; *Okada et al., 2013*). Active Cdc42 recruits septins to the polarization site. The septin ring that is formed by polarized exocytosis corrals exocyst-dependent exocytosis and active Cdc42 inside the ring (*Dobbelaere and Barral, 2004*; *Okada et al., 2013*). However, there is no evidence that the exocyst and septins physically and directly interact in budding yeast. Consistently, recent mapped *S. cerevisiae* protein interactome found no interactions between septins and exocyst in the pull-down experiments (*Michaelis et al., 2023*).

By contrast, several interactions between septins and the exocyst have been identified by Co-IPs to support the role of septins in the regulation of the exocyst localization in other cell types (*Hsu et al., 1998*; *Beites et al., 1999*; *Vega and Hsu, 2003*; *Li et al., 2007*; *Gupta et al., 2015*). During hyphal development in *C. albicans*, association of Sec3 and Sec5 with the septin Cdc3 was detected by Co-IP (*Li et al., 2007*). In *M. oryzae*, mislocalization of Sec6 was reported with deletion of the septin Sep3. This was supported by pull-down and mass spectrometry data where Sep4 and Sep5 were pulled down by Exo84 while Sep3 was pulled down by Sec6 (*Gupta et al., 2015*). Interactions between septins and the exocyst in our study are also consistent with the earlier study in rat brain where the exocyst and septins are found to directly interact with moderate affinity (*Hsu et al., 1998*). Using rat brain lysates, septins were co-purified by the anti-Sec8 antibody, and purified septins and the exocyst complex Co-IP with each other (*Hsu et al., 1998*). Consistently, the exocyst detected using anti-Sec6 antibody shows partial colocalization with the septin CDC10/Septin 7 in cultured hippocampal neurons (*Hsu et al., 1998*). Moreover, the exocyst subunits Sec8 and Exo70, along with tubulin, co-immunoprecipitated with the septin Nedd5 from rat brain cells (*Vega and Hsu, 2003*). Therefore, the apparent absence of an interaction between septins and the exocyst in budding yeast may be an outlier when it comes to conservation of this interaction. Further analyses of these complexes in other model systems are needed to confirm this hypothesis. Here, we have presented comprehensive studies on explaining the importance of septins in regulating exocytosis by likely direct physical interactions with the exocyst in fission yeast.

In our study, we systematically investigated all the potential pairwise interactions between septin and exocyst subunits using AlphaFold2 predictions. We experimentally confirmed five out of the six predicted interactions by Co-IPs: Spn1–Sec15, Spn1–Sec6, Spn2–Sec15, Spn2–Sec5, and Spn4–Sec15

and validated four of them by yeast two-hybrid assays (except Spn2–Sec5 due to high levels of Sec15 autoactivation). The observed associations are consistent with direct interactions predicted by Alpha-Fold2 but cannot alone establish their direct bindings. These multivalent interactions ensure that the exocyst dynamically tethers secretory vesicles on the plasma membrane with high temporal and spatial fidelity, even if individual interactions may not be very strong. The subunits Sec15, Sec6, and Sec5 in the exocyst complex are known to be available for interacting with many proteins as mentioned above in budding yeast and in other systems for different cellular functions (*Sjölinder et al., 2002*; *Fukai et al., 2003*; *Zhang et al., 2004*; *Feng et al., 2012*; *Du et al., 2015*; *Guo et al., 2016*; *Yang et al., 2022*). We also predicted septin octameric and hexameric complexes and the *S. pombe* exocyst structure using AlphaFold3. The exocyst structure is most likely based on the published structural models of the full exocyst complex from budding yeast (e.g., PDB: 5YFP; *Lepore et al., 2018*; *Mei et al., 2018*). We found that the majority of the predicted exocyst–septin-interacting residues are located on the accessible surfaces of the assembled whole complexes (*Figure 3—figure supplement 2*; *Figure 3—figure supplement 3*; *Figure 3—video 1*; *Figure 3—video 2*; *Figure 3—video 3*; *Figure 3—video 4*). These predictions indicate that these septin–exocyst interactions are sterically plausible. The interactions between septins and the exocyst that we identified in fission yeast will provide important insights into the mechanisms of exocyst regulations by septins. During evolution, fission yeast may have lost many but some of the most conserved aspects of septin functions, including the septin–exocyst interactions.

It is known that the octameric exocyst complex consists of two subcomplexes (*Heider et al., 2016*; *Ahmed et al., 2018*; *Lepore et al., 2018*; *Mei and Guo, 2018*; *Mei et al., 2018*; *Ganesan et al., 2020*). Subcomplex 1 consists of Sec3, Sec5, Sec6, and Sec8 while subcomplex 2 consists of Sec10, Sec15, Exo70, and Exo84. In our study, we found that septins can interact with both exocyst subcomplexes with multivalent interactions by AlphaFold predictions, reciprocal Co-IPs, and yeast two-hybrid assays. Some of the identified interactions may only be strong enough between specific subunits at exposed interfaces under the Co-IP conditions, rather than through the whole complex as predicted by AlphaFold. Additionally, the detergent and salt conditions used in our Co-IPs may disrupt labile complex interfaces or partially dissociate multimeric assemblies. Future studies are needed to refine the residues involved in the interactions because the predicted interacting residues from AlphaFold are too numerous. However, it is encouraging that most of the predicted interacting residues are clustered in several surface patches. Experimental validation through targeted mutagenesis is an important next step. In addition, tests are needed to figure out if posttranslational modifications are necessary for the interactions between septins and the exocyst. Because the colocalization of septins and the exocyst required for their proper function occurs at specific stages during cytokinesis rather than a general regulation throughout the cell cycle, septin filament formation and posttranslational modifications of the involved proteins are most likely required, which make it challenging to tease out the interactions in vitro (*Dobbelaere et al., 2003*; *Hernández-Rodríguez and Momany, 2012*; *Ren and Guo, 2012*; *Tay et al., 2019*; *Sharma and Menon, 2023*; *Werner and Yadav, 2023*). Moreover, we cannot rule out that Rho1/RhoA GTPase and PI(4,5)P2 are involved in septin–exocyst interactions as both have been reported to interact with septins and/or the exocyst in other cell types (*Guo et al., 2001*; *He et al., 2007*; *Bertin et al., 2010*; *Bendezú and Martin, 2011*; *Pérez et al., 2015*; *Carim and Hickson, 2023*; *Safavian et al., 2023*).

In summary, we found that septins are important for exocyst targeting to the division site during cytokinesis through multivalent interactions between their subunits. The proper exocyst localization at the rim of the division plane is critical for timely and successful cytokinesis. Our results will provide insights into future studies of the interactions and functions of both septins and the exocyst in other cell types. Dysregulation of septins or the exocyst leads to severe disorders including neurological diseases and cancers (*Russell and Hall, 2005*; *Martin-Urdiroz et al., 2016*; *Halim et al., 2023*; *Werner and Yadav, 2023*). Thus, it is important to identify the functional and physical links between septins and the exocyst complex in human cells.

## Materials and methods
### Strains and molecular methods
Fission yeast strains used in this study are listed in *Supplementary file 1*. Strains were constructed using PCR-based gene targeting and standard genetic methods (*Moreno et al., 1991*; *Bähler et al., 1998*).

DNA oligos used in this study are provided in **Supplementary file 2**. Tagged genes were expressed under endogenous promoters and integrated at their native chromosomal loci except where noted. The glucan synthase gene *bgs1* is integrated at the *leu1* loci under endogenous promoter, with the endogenous copy deleted (*Cortés et al., 2002*). The functionalities of the newly tagged proteins (Spn1, Spn2, Spn4, Sec3, Sec5, Sec6, Sec8, Sec15, and Exo70) were tested by growing the strains at 25 and 36°C on YE5S media or crossing to mutants. The growth and morphology of the tagged strains were comparable to WT.

## Microscopy

Cells were normally grown at the exponential phase in YE5S liquid medium at 25°C for 40–48 hr before microscopy or temperature shift. Confocal microscopy was performed as previously described (*Wang et al., 2014*; *Davidson et al., 2015*; *Davidson et al., 2016*; *Zhu et al., 2018*). Briefly, cells were collected from liquid culture by centrifuging at 3000 rpm for 30 s at room temperature and washed with EMM5S twice to reduce autofluorescence. A final concentration of 5 μM *n*-propyl-gallate (*n*-PG) from a 10x stock (in EMM5S) was added in the second wash to protect cells from free radicals during imaging. Live cells were imaged on a thin layer of EMM5S with 20% gelatin and 5 μM *n*-PG at ~23°C. To image cells at 36°C, concentrated cells were spotted into coverglass-bottom dish and covered with EMM5S agar (*Davidson et al., 2016*).

We imaged cells using several microscopy systems with 100x/1.4 or 100x/1.45 numerical aperture (NA) Plan-Apo objective lenses (Nikon, Melville, NY). Most fluorescence images were taken using a PerkinElmer spinning disk confocal system (UltraVIEW Vox CSUX1 system; PerkinElmer, Waltham, MA) with 440-, 488-, 515-, and 561-nm solid-state lasers and back-thinned electron-multiplying charge-coupled device (EMCCD) cameras (C9100-13 or C9100-23B; Hamamatsu Photonics, Bridgewater, NJ) on a Nikon Ti-E inverted microscope. For better spatial resolution, **Figure 1A** was imaged using another spinning disk confocal system (UltraVIEW ERS; PerkinElmer) with 568 nm solid-state laser and 488 nm argon ion lasers and a cooled charge-coupled device camera without binning (ORCA-AG; Hamamatsu Photonics) on a Nikon Eclipse TE2000-U microscope. For the SoRa imaging shown in **Figure 1C**, the images were captured with a Nikon CSU-W1 SoRa spinning disk confocal system equipped with 488 and 561 nm solid-state lasers and an ORCA-Quest qCMOS camera (C15550, Hamamatsu Photonics, Bridgewater, NJ) on a Nikon Eclipse Ti-2E microscope with 2 × 2 binning (**Ye et al., 2025**). We used TIRF microscopy controlled by NIS Elements software to examine the dynamic localization of the exocyst subunit Exo70 and the septin Spn1 at the division site for some movies (**Figure 1—video 1**; **Figure 1—video 2**; **Figure 1—video 3**). A Nikon Eclipse Ti-E microscope equipped with a TIRF illuminator, Plan Apo 100x/1.45NA oil objective, and an Andor iXon Ultra 897 EMCCD was used.

## Image analysis

We analyzed images using ImageJ/Fiji (National Institutes of Health, Bethesda, MD) and Volocity (PerkinElmer). Fluorescence images are maximum-intensity projections from z-sections spaced at 0.5 μm except where noted. Images of 3D projections (end-on views) and deconvolution (**Figure 7C**, Eng1) were generated from images with z-sections spaced at 0.05 μm. For quantification of fluorescence intensity at the division site, we summed the intensity from all z-sections using sum projection. A rectangular ROI1 was drawn to include the majority of division site signal for intensity measurement. Then the intensity in a second ROI2 approximately twice the area of ROI1 (including ROI1) was measured and used to subtract cytoplasmic background as described previously (*Coffman et al., 2011*; *Davidson et al., 2015*; *Davidson et al., 2016*).

For comparing the colocalization at the rim of the division plane (**Figure 1B**), a line along the cell long-axis was drawn across the division plane at the same position for both Spn1 and Sec3 channels using maximum intensity projection images. Then the width of the line was adjusted to cover all signals at the division site, generating an ROI of 1.5 μm × 3.5 μm (*x–y*) (see **Figure 1B**). The mean intensity of all pixels in the *y*-axis was measured along the *x*-axis and plotted.

Line scans (**Figures 6D and 7A, C**) across the division plane were made in the middle focal plane of the fluorescence images. A line along the cell short axis was drawn across the division plane of the cells with a closed septum (at or after the end of contractile-ring constriction) to cover the whole cell diameter. To quantify their fluorescence intensity at the division site using line scans, the line width

used was 3 pixels to reduce signal variations caused by measurements on a single-focal plane. Mean intensity (average of 3 pixels) was measured across cell diameter. For Syb1 (*Figure 6D*), cells at the end of ring constriction (indicated by an Rlc1 dot at the center of the division plane) were measured; and line scans were aligned by referencing the peak intensity of Rlc1 signal. All data were aligned by the center and plotted. For Bgs1 (*Figure 7A*), we quantified the cells from which the Rlc1 signal had disappeared from the division site. The line was drawn in the Bgs1 channel in the middle focal plane. The center of line scan was defined as the pixel with the brightest Bgs1 value. All data were aligned by the center and plotted. For Eng1 (*Figure 7C*), cells with closed septa were measured, and line scans for WT cells were aligned by the middle of the two peaks, and the ones for *spn1Δ* cells were aligned by referencing the middle of septa in DIC images.

## FRAP analysis

FRAP was performed using the photokinesis unit on the UltraVIEW Vox confocal system at either ~23 or 36°C (*Coffman et al., 2009*; *Laporte et al., 2011*; *Zhu et al., 2013*). Half of the division site signals at the middle focal plane were photobleached to <50% of the original fluorescence intensity. Five pre-bleach images and 150 post-bleach images for *spn1Δ* cells, or 70 post-bleach images for *sec3-913* cells, were collected at every 0.33 or 10 s, respectively. For image analysis, the background and photo-bleaching during image acquisition were corrected using empty space and unbleached cells within the same image. The pre-bleach intensity was normalized to 100%, and the first post-bleach intensity was normalized to 0% (*Laporte et al., 2011*; *Zhu et al., 2018*). Intensities of three consecutive post-bleach time points were rolling averaged to reduce noise (*Vavylonis et al., 2008*). Data were plotted and fitted using the exponential decay equation $y = m_1 + m_2 \exp(-m_3 x)$, where $m_3$ is the off-rate. The half-time for recovery was calculated by $t_{1/2} = \ln 2/m_3$.

## Predictions of septin–exocyst interactions using AlphaFold analyses

The development of computer algorithms to predict three-dimensional protein structures from amino acid sequence involves two complementary ways that concentrate on either the physical interactions or the evolutionary history (*Jumper et al., 2021*). AlphaFold utilizes cutting-edge neural network topologies and training techniques to predict the 3D coordinates of a primary amino acid sequence (*Jumper et al., 2021*). We made the AlphaFold models of interactions between different septin and exocyst subunits using Google Colab Platform and AlphaFold2_advanced option that does not need templates at https://colab.research.google.com/github/sokrypton/ColabFold/blob/main/beta/AlphaFold2_advanced.ipynb#scrollTo=ITcPnLkLuDDE. Sequences of each subunit were searched against genetic databases with msa_method = mmseqs2, pair_mode = unpaired. The default mode of sampling options was used: num_models = 5, ptm option, num_ensemble = 1, max_cycles = 3, num_samples = 1. A total of five models were ranked according to their pLDDT score between 0 and 100, from low to high confidence level. Septin and exocyst subunits were input in a 1:1 ratio. For each of the 32 pairs of septin and exocyst subunits, the protein sequences were entered in both orders (e.g., Spn1:Sec3 and Sec3:Spn1). We found that the order of input sequence affects some prediction results. So we predicted all septin–exocyst combinations in both input sequence orders. We then selected the top septin–exocyst combinations that showed interactions in both input orders. The structural figures were drawn with PyMOL version 2.0 (Schrodinger, Inc).

We used AlphaFold3 (https://alphafoldserver.com/) to predict the structures of fission yeast exocyst complex and septin hexamer/octamer (*Abramson et al., 2024*). To predict the structure of the whole exocyst complex, we trimmed some of the exocyst subunits to meet the 5000-residue limit of AlphaFold3 based on the budding yeast cryo-EM structure of exocyst complex (PDB: 5YFP, 4.4 Å resolution) (*Mei et al., 2018*). The truncations were selected so that they do not interfere with inter-subunit interactions as well as with septin binding based on our modeling. Sequences of all subunits of the respective complexes were used as input and models were generated using the default settings. Top-ranked models based on PAE, pTM, and iPTM were analyzed in PyMol. Different subunits were colored distinctly to differentiate the interface. To evaluate the accessibility of residues, surface exposure of the predicted interacting residues was mapped onto the corresponding residues in the final models and colored yellow to visualize distinctly.

## Co-IP and western blotting

We carried out Co-IP and western blotting as previously described (*Laporte et al., 2011*; *Lee and Wu, 2012*; *Ye et al., 2012*). Briefly, mEGFP, GFP, mYFP, or 13Myc-tagged septin or exocyst subunits were expressed under native promoters in fission yeast. Cells were grown in YE5S liquid medium at 25°C for ~48 hr before harvesting and lyophilization. Lyophilized cells (200 mg) were ground into a homogeneous fine powder using pestles and mortars. IP buffer (50 mM 4-(2-hydroxyethyl)-1-piperaz ineethanesulfonic acid [HEPES], pH 7.5, 150 mM NaCl, 1 mM EDTA, 0.1% NP40, 50 mM NaF, 20 mM glycerophosphate, 0.1 mM $Na_3VO_4$, 1 mM PMSF, and protease inhibitor [11873580001, Roche] 1 tablet/30 ml buffer) was added according to the ratio of 10 µl: 1 mg lyophilized cell powder. 60 µl Dynabeads protein G beads (10004D, Invitrogen) were incubated with 5 µg polyclonal GFP antibody (NB600-308, Novus Bio) for 1 hr at room temperature. After three washes with PBS and one wash with 1 ml IP buffer, the beads were incubated with cell lysate for 2 hr at 4°C. After five washes at 4°C with 1 ml IP buffer each time, proteins were eluted by boiling with 80 µl sample buffer. The protein samples were separated with SDS–PAGE gel and detected with monoclonal anti-GFP antibody (1:1000 dilution; 11814460001; Roche, Mannheim, Germany), monoclonal anti-Myc antibody (1:500 dilution, 9E10, Santa Cruz Biotechnology, Dallas, TX), and anti-tubulin TAT1 antibody at 1:10,000 dilution (*Woods et al., 1989*). Secondary antibody anti-mouse immunoglobulin G (1:5000 dilution; A4416, Sigma-Aldrich) was detected using SuperSignal Maximum Sensitivity Substrate (34096, Thermo Fisher Scientific) on iBright CL1500 imager (Thermo Fisher Scientific).

## Yeast two-hybrid assays

Yeast two-hybrid assays were performed as described previously using X-gal overlay and β-D-galactosidase activity quantifications (*Amberg et al., 2006*; *Paiano et al., 2019*). DNA or cDNA (for genes with introns) sequences of Spn1, Spn1(aa 300–469), Spn2, Spn4, Sec5, Sec6, and Sec15 were cloned into pVP16 or pGBT9 vectors having VP16 transcription activation domain (AD) or GAL4 transcription factor DNA-binding domain (BD), respectively. Constructed plasmids were confirmed by restriction digestions and Sanger sequencing. Pairs of plasmids were then co-transformed into *S. cerevisiae* strain MAV203 (11281-011; Invitrogen) and plated on synthetic drop-out medium lacking leucine and tryptophan (SD-L-W) for selection. For X-gal overlay assay, grown colonies were re-streaked on YPD (yeast extract-peptone-dextrose) plates to grow overnight. We used 10–12 ml chloroform per plate to permeabilize cells for 10 min and then dried for 10 additional min. 0.5% agarose was prepared in 25 ml PBS (pH 7.5) and 500 µl X-gal (20 mg/ml stock in DMSO) was added after cooling. After mixing thoroughly, agarose containing X-gal was overlaid onto the colonies and incubated at 30°C. Plates were checked every 30 min for development of blue color.

Interactions were then quantified by β-D-galactosidase activity using the ONPG assay (48712-M; Sigma-Aldrich) according to the published methods (*Amberg et al., 2006*; *Paiano et al., 2019*). For interactions between Sec15 with Spn1, Spn2, and Spn4, the Amberg et al. method was used (*Amberg et al., 2006*). Briefly, cells were grown in SD-L-W liquid medium at 30°C overnight. 40 ml culture with $OD_{595}$ >1 was collected and washed with 1 ml distilled water. Then cells were broken in 110 µl breaking buffer (100 mM Tris-Cl, pH 7.5, 1 mM DTT, and 20% glycerol) using glass beads on bead beater. 10 µl of the lysate was diluted with 90 µl distilled water and spun down to remove cell debris, and the supernatant was used to estimate protein concentration by Bradford assay. To the remaining 100 µl of lysate, 0.9 ml Z-buffer (100 mM sodium phosphate, pH 7.5, 10 mM KCl, and 2 mM $MgSO_4$) and 0.2 ml ONPG (8 mg/1 ml Z buffer) were added and incubated at 28°C until pale yellow color developed in at least one of the samples. All the reactions were stopped by adding 0.4 ml 1 M $Na_2CO_3$. Debris was removed by centrifuging at 15,700 × *g* for 10 min and $OD_{420}$ was measured using 1 ml of supernatant. Time elapsed from adding ONPG to adding stop solution was recorded and activity of β-galactosidase was calculated using the formula:

β-galactosidase activity (nmol/min/mg) = $OD_{420}$ × 1.7/[0.0045 × protein (mg/ml) × extract volume (ml) × time (min)].

For the interaction between Spn1 and Sec6, the Paiano et al. method was used (*Paiano et al., 2019*). Briefly, cultures were diluted to $OD_{595}$ = 0.30 and incubated for 2 hr at 30°C. For each sample, cells from 9 ml culture were collected and washed with 1 ml Z buffer and then resuspended in 0.1 ml Z buffer. Cells were broken by three freeze–thaw cycles in liquid nitrogen. 0.7 ml Z buffer with β-mercaptoethanol (27 µl β-mercaptoethanol in 9.973 ml Z buffer) and 160 µl ONPG was added to the cell

lysates and incubated at 30°C until a yellow color developed in at least one of the samples. Reactions were stopped by adding 0.4 ml 1 M $Na_2CO_3$. Debris was removed by centrifuging at 15,700 × $g$ for 10 min and $OD_{420}$ was measured using 1 ml supernatant. Time elapsed from adding ONPG to adding stop solution was recorded and β-galactosidase activity was calculated using the following formula:

β-galactosidase units = $1000 \times OD_{420}/[T \times V \times OD_{595}]$, where $T$ is the elapsed time (min), $V$ is the volume (ml) of culture used, and $OD_{595}$ is the optical density of yeast culture.

### Electron microscopy

Electron microscopy was performed at the Boulder Electron Microscopy Services at the University of Colorado, Boulder (Boulder, CO) as previously described (*Lee et al., 2014*; *Wang et al., 2016*). Briefly, yeast cells were grown at 25°C for ~41 hr in YE5S medium and then shifted to 36°C for 4 hr before harvesting using Millipore filters. Samples were prepared using high-pressure freezing with a Wohlwend Compact 02 Freezer in the presence of 2% osmium tetroxide and 0.1% uranyl acetate in acetone. Thin sections with a thickness of 70 nm were cut and embedded in Epon-Araldite epoxy resin, which was post-stained with uranyl acetate and lead citrate. Imaging of EM samples was done using a Philips CM100 transmission electron microscope (FEI, Hillsboro, OR).

### Statistical analysis

Data in graphs are mean ± SD except where noted. The p-values in statistical analyses were calculated using the two-tailed Student's $t$ tests except *Figure 4E, F*, where one-way ANOVA with Tukey's post hoc test was used to quantify yeast two-hybrid analyses.

## Acknowledgements

We thank Mohan Balasubramanian, Sophie Martin, Pilar Pérez, John Pringle, and Takashi Toda for fission yeast strains; Eileen O'Toole and Garry Morgan at The University of Colorado, Boulder, for help with electron microscopy; Anita Hopper, Steve Osmani, Dmitri Kudryashov, Elena Kudryashova, Damien Wilburn, and Emily Vais for equipment and technical support; and members of the Wu laboratory for helpful discussion and suggestions. This study was funded by Pelotonia Graduate Fellowship to Yajun Liu, Pelotonia Undergraduate Fellowship to Shelby Naegele, and the National Institute of General Medical Sciences of NIH grant GM118746 to Jian-Qiu Wu. The funders had no role in study design, data collection, and interpretation, or the decision to submit the work for publication.

## Additional information

### Funding

| Funder | Grant reference number | Author |
| --- | --- | --- |
| National Institutes of Health | GM118746 | Jian-Qiu Wu |
| The Ohio State University | Pelotonia Graduate Fellowship | Yajun Liu |
| The Ohio State University | Pelotonia Undergraduate Fellowship | Shelby M Naegele |

The funders had no role in study design, data collection, and interpretation, or the decision to submit the work for publication.

### Author contributions

Davinder Singh, Conceptualization, Data curation, Formal analysis, Validation, Investigation, Visualization, Methodology, Writing – original draft, Writing – review and editing; Yajun Liu, Conceptualization, Data curation, Formal analysis, Funding acquisition, Validation, Investigation, Methodology, Writing – original draft, Writing – review and editing; Yi-Hua Zhu, Sha Zhang, Investigation, Visualization, Methodology, Writing – review and editing; Shelby M Naegele, Funding acquisition, Investigation, Writing – review and editing; Jian-Qiu Wu, Conceptualization, Resources, Supervision, Funding

acquisition, Validation, Investigation, Methodology, Writing – original draft, Project administration, Writing – review and editing

### Author ORCIDs
Davinder Singh 
Yi-Hua Zhu 
Jian-Qiu Wu 

Reviewer #1 (Public review): https://doi.org/10.7554/eLife.101113.3.sa1
Reviewer #2 (Public review): https://doi.org/10.7554/eLife.101113.3.sa2
Author response https://doi.org/10.7554/eLife.101113.3.sa3

## Additional files

### Supplementary files
Supplementary file 1. *S. pombe* strains used in this study.

Supplementary file 2. DNA oligos used in this study.

MDAR checklist

### Data availability
All data are available in the main text, the supplementary materials, or the source files.

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
