## [Editor Report · eLife Assessment]

How secretion is regulated during cell division and how membrane trafficking factors cooperate with the cytoskeleton during cell division remain poorly understood. In this work the authors find protein-protein interactions and localization dependencies between the polymeric septin cytoskeleton and the exocyst complex, using fission yeast as a model organism and using alphafold 3 based structural predictions. The work provides a **valuable** body of new information that will be of great interest to the cell biology community. The evidence is **solid** and provides the authors and the community a framework to test if the identified interfaces reflect bona fide interaction sites in vivo and in vitro in future.

---

## [Referee Report · Reviewer #1 (Public review)]

Summary

In this manuscript, Singh, Wu and colleagues explore functional links between septins and the exocyst complex. The exocyst in a conserved octameric complex that mediates the tethering of secretory vesicles for exocytosis in eukaryotes. In fission yeast cells, the exocyst is necessary for cell division, where it localizes mostly at the rim of the division plane, but septins, which localize in a similar manner, are non-essential. The main findings of the work are that septins are required for the specific localization of the exocyst to the rim of the division plane, and the likely consequent localization of the glucanase Eng1 at this same location, where it is known to promote cell separation. In absence of septins, the exocyst still localizes to the division plane, but is not restricted to the rim. They also show some defect in the localization of secretory vesicles and glucan synthase cargo. They further show interactions between septins and exocyst subunits through coIP experiments.

Strengths

The septin, exocyst and Eng1 localization data are well supported, showing that the septin rim recruits the exocyst and (likely consequently) the Eng1 glucanase at this location. One important finding of the manuscript is that of a physical interaction between septins and exocyst subunits in co-immunoprecipitation experiments.

Weaknesses

While interactions are supported by coIP experiments, the AlphaFold-predicted septin-exocyst interactions are not very convincing and the predicted binding interfaces are not supported by mutation analysis. A further open question is whether septins interact with the intact exocyst complex or whether the interactions occur only with individual subunits. The two-hybrid and coIP data only show weak interactions with individual subunits, and some coIPs (for instance Sec3 and Exo70 with Spn1 and Spn4) are negative, suggesting that the exocyst complex may not remain intact in these experiments.

---

## [Referee Report · Reviewer #2 (Public review)]

Summary:

This interesting study implicates the direct interaction between two multi-subunit complexes, known as the exocyst and septin complexes, in the function of both complexes during cytokinesis in fission yeast. While previous work from several labs had implicated roles for the exocyst and septin complexes in cytokinesis and cell separation, this study describes the importance of protein:protein interaction between these complexes in mediating the functions of these complexes in cytokinesis. Previous studies in neurons had suggested interactions between septins and exocyst complexes occur but the functional importance of such interactions was not known. Moreover, in baker's yeast where both of these complexes have been extensively studied - no evidence of such an interaction has been uncovered despite numerous studies which should have detected it. Therefore while exocyst:septin interactions appear to be conserved in several systems, it appears likely that budding yeast are the exception--having lost this conserved interaction.

Strengths:

The strengths of this work include the rigorous analysis of the interaction using multiple methods including Co-IP of tagged but endogenously expressed proteins, 2 hybrid interaction, and Alphafold Multimer. Careful quantitative analysis of the effects of loss of function in each complex and the effects on localization and dynamics of each complex was also a strength. Taken together this work convincingly describes that these two complexes do interact and that this interaction plays an important role in post Golgi vesicle targeting during cytokinesis.

Comments on revisions:

The authors have added substantial work to the revised manuscript, and it is much improved. In particular, the figures portraying the AlphaFold Multimer model of the exocyst:septin interactions are much clearer. I also appreciate the effort that went into modeling the fission yeast exocyst complex based on the yeast CryoEM structure in order to determine if the predicted interfaces with septins were likely to be surface accessible in the intact exocyst complex.

---

## [Author Response]

The following is the authors’ response to the original reviews.

**Reviewer #1 (Public Review):**
Summary:In this manuscript, Singh, Wu and colleagues explore functional links between septins and the exocyst complex. The exocyst in a conserved octameric complex that mediates the tethering of secretory vesicles for exocytosis in eukaryotes. In fission yeast cells, the exocyst is necessary for cell division, where it localizes mostly at the rim of the division plane, but septins, which localize in a similar manner, are non-essential. The main findings of the work are that septins are required for the specific localization of the exocyst to the rim of the division plane, and the likely consequent localization of the glucanase Eng1 at this same location, where it is known to promote cell separation. In the absence of septins, the exocyst still localizes to the division plane but is not restricted to the rim. They also show some defects in the localization of secretory vesicles and glucan synthase cargo. They further propose that interactions between septins and exocysts are direct, as shown through Alphafold2 predictions (of unclear strength) and clean coIP experiments.Strengths:The septin, exocyst and Eng1 localization data are well supported, showing that the septin rim recruits the exocyst and (likely consequently) the Eng1 glucanase at this location. One major finding of the manuscript is that of a physical interaction between septins and exocyst subunits. Indeed, many of the coIPs supporting this discovery are very clear.Weaknesses:I am less convinced by the strength of the physical interaction of septins with the exocyst complex. Notably, one important open question is whether septins interact with the intact exocyst complex, as claimed in the text, or whether the interactions occur only with individual subunits. The two-hybrid and coIP data only show weak interactions with individual subunits, and some coIPs (for instance Sec3 and Exo70 with Spn1 and Spn4) are negative, suggesting that the exocyst complex does not remain intact in these experiments.Given the known structure of the full exocyst complex and septin filaments (at least in *S. cerevisiae*), the Alphafold2 predicted structure could be used to probe whether the proposed interaction sites are compatible with full complex formation.

We thank the reviewer for these important and insightful comments. We agree that our current data, particularly the data from yeast two-hybrid and co-immunoprecipitation (coIP) assays, primarily reveal interactions between individual septin and exocyst subunits, and do not conclusively demonstrate binding of septins to the fully assembled exocyst complex. We realize this as a key limitation and have revised the manuscript text accordingly to clarify this point.

We also appreciate the reviewer’s suggestion to use structural prediction to further assess their interaction plausibility. We have now employed the full *Saccharomyces cerevisiae* exocyst complex (with 4.4 Å resolution) published by the Guo group (Mei et al., 2018) to examine the interfaces of septin and the exocyst interactions, assuming that the *S. pombe* exocyst has the similar structure. We focused on checking all the interacting residues on the exocyst complex and septins from our AlphaFold modeling to determine whether these predicted interactions are structurally compatible. Our analysis reveals that majority subunit interactions are sterically feasible, while a few would likely require partial disassembly or flexible conformations. These new insights have been added to the revised Results and Discussion sections (Figure Supplement S4, S5 and Videos 4-7).

While we cannot fully resolve whether septins engage with the whole exocyst complex versus selected subunits, our combined data support a model that septins scaffold or spatially regulate the exocyst localization at the division site, potentially through dynamic and multivalent interactions. We now explicitly state this more cautious interpretation in the revised manuscript.

Mei, K., Li, Y., Wang, S., Shao, G., Wang, J., Ding, Y., Luo, G., Yue, P., Liu, J.-J., Wang, X. and Dong, M.-Q., Wang, H-W, Guo W. 2018. Cryo-EM structure of the exocyst complex. Nature Struct & Mol. Biol, 25(2), pp.139-146.

The effect of spn1∆ on Eng1 localization is very clear, but the effect on secretory vesicles (Ypt3, Syb1) and glucan synthase Bgs1 is less convincing. The effect is small, and it is not clear how the cells are matched for the stage of cytokinesis.

For localizations and quantifications of Eng1, Ypt3, Syb1, and Bgs1 shown in Figures 6 and 7, cells with a closed septum (at or after the end of contractile-ring constriction) were quantified or highlighted. To quantify their fluorescence intensity at the division site using line scan, the line width used was 3 pixels. For Syb1 (Figure 6D), we quantified cells at the end of ring constriction (when Rlc1-tdTomato constricted to a dot) in the middle focal plane. The exact same lines were drawn in both Rlc1 and Syb1 channels. The center of line scan was defined as the pixel with the brightest Rlc1 value. All data were aligned by the center and plotted. For Bgs1 (Figure 7A), we quantified the cells that Rlc1 signal had disappeared from the division site. The line was drawn in the Bgs1 channel in the middle focal plane. The center of line scan was defined as the pixel with the brightest Bgs1 value.

All data were aligned by the center and plotted. These details were added to the Materials and Methods.

**Reviewer #2 (Public Review):**
Summary:This interesting study implicates the direct interaction between two multi-subunit complexes, known as the exocyst and septin complexes, in the function of both complexes during cytokinesis in fission yeast. While previous work from several labs had implicated roles for the exocyst and septin complexes in cytokinesis and cell separation, this study describes the importance of protein:protein interaction between these complexes in mediating the functions of these complexes in cytokinesis. Previous studies in neurons had suggested interactions between septins and exocyst complexes occur but the functional importance of such interactions was not known. Moreover, in baker's yeast where both of these complexes have been extensively studied - no evidence of such an interaction has been uncovered despite numerous studies which should have detected it. Therefore while exocyst:septin interactions appear to be conserved in several systems, it appears likely that budding yeast are the exception--having lost this conserved interaction.Strengths:The strengths of this work include the rigorous analysis of the interaction using multiple methods including Co-IP of tagged but endogenously expressed proteins, 2 hybrid interaction, and Alphafold Multimer. Careful quantitative analysis of the effects of loss of function in each complex and the effects on localization and dynamics of each complex was also a strength. Taken together this work convincingly describes that these two complexes do interact and that this interaction plays an important role in post Golgi vesicle targeting during cytokinesis.Weaknesses:The authors used Alphafold Multimer to predict (largely successfully) which subunits were most likely to be involved in direct interactions between the complexes. It would be very interesting to compare this to a parallel analysis on the budding yeast septin and exocyst complexes where it is quite clear that detectable interactions between the exocyst and septins (using the same methods) do not exist. Presumably the resulting pLDDT scores will be significantly lower. These are in silico experiments and should not be difficult to carry out.

We thank the reviewer for this insightful suggestion. To assess the specificity of the predicted interactions between septins and the exocyst complex in *S. pombe*, we performed a comparative AlphaFold2 analysis using some of the homologous subunits from *Saccharomyces cerevisiae*. We modeled two interactions between Cdc10-Sec5 and Cdc10-Sec15 (Cdc10 is the Spn2 homolog) using the same pipeline and parameters at the time when we did the modeling for *S. pombe*. We did not find interactions between them using the criteria we used for the fission yeast proteins in this study. These results support the notion that the predicted septin–exocyst interactions in *S. pombe* are not generalizable to budding yeast. Unfortunately, we did not test all other combinations at that time and the AlphaFold2 platform is not available to us now (showing system error messages when we tried recently). We thank the reviewer again for this helpful suggestion, which should strengthen the evolutionary interpretation of the septin-exocyst interactions once it is able to be systematically carried out.

**Reviewer #3 (Public Review):**
Septins in several systems are thought to guide the location of exocytosis, and they have been found to interact with the exocyst vesicle-tethering complex in some cells. However, it is not known whether such interactions are direct or indirect. Moreover, septin-exocyst physical associations were not detected in several other systems, including yeasts, making it unclear whether such interactions reflect a conserved septin-exocytosis link or whether they may missed if they depend on septin polymerization or association into higher-order structures. Singh et. al., set out to define whether and how septins influence the exocyst during *S. pombe* cytokinesis. Based on three lines of evidence, the authors conclude that septins directly bind to exocyst subunits to regulate localization of the exocyst and vesicle secretion during cytokinesis. The conclusions are consistent with the data presented, but some interpretations need to be clarified and extended:(1) The first line of evidence examines septin and exocyst localization during cytokinesis in wild-type and septin-mutant or exocyst-mutant yeast. Quantitative imaging convincingly shows that the detailed localization of the exocyst at the division site is perturbed in septin mutants, and that this is accompanied by modest accumulation of vesicles and vesicle cargos. Whether that is sufficient to explain the increased thickness of the division septum in septin mutants remains unclear.

The modest accumulation of vesicles and vesicle cargos at the division site is one of the reasons for the increased thickness of the division septum in septin mutants. It is more likely that the misplaced exocyst can still tether vesicles along the division plane (less likely at the rim) without septins. Due to the lack of the glucanase Eng1 at the rim of the division plane in septin mutants, daughter-cell separation is delayed and then cells continue to thicken the septum. We have added these points to the Discussion.

(2) The second line of evidence involves a comprehensive Alphafold2 analysis of potential pair-wise interactions between septin and exocyst subunits. This identifies several putative interactions in silico, but it is unclear whether the identified interaction surfaces would be available in the full septin or exocyst complexes.

We thank the reviewer for raising this important point. We fully agree that a key limitation of pairwise AlphaFold predictions is that they do not account for the higher-order structural context of multimeric protein complexes, such as septin hetero-oligomers or the assembled exocyst complex. As a result, some of the predicted interfaces could indeed be conformationally restricted in the native state.

To address this concern, we predicted the *S. pombe* exocyst and septin structures using AlphaFold3. We mapped predicted contact residues onto the predicted structure. Most predicted interfaces (86% for the exocyst and 86-96% for septins) appear to be located on accessible surfaces in the assembled complexes (Figure supplement S4, S5, videos 4 - video 7), suggesting that these interactions are sterically plausible. We have added this important caveat to the text of the revised manuscript highlighting the interface accessibility within the assembled complexes. We appreciate the reviewer’s insight, which helped us strengthen the interpretation and limitations of the AlphaFold-based analysis.

(3) The third line of evidence uses co-immunoprecipitation and yeast two hybrid assays to show that several physical interactions predicted by Alphafold2 can be detected, leading the authors to conclude that they have identified direct interactions. However, both methods leave open the possibility that the interactions are indirect and mediated by other proteins in the fission yeast extract (co-IP) or budding yeast cell (two-hybrid).

We thank the reviewer for this important clarification. We agree that coimmunoprecipitation (co-IP) and yeast two-hybrid (Y2H) assays cannot conclusively distinguish between direct and indirect interactions. As the reviewer points out, co-IPs may reflect associations mediated by bridging proteins within the fission yeast extract, and Y2H readouts can be influenced by fusion context or endogenous host proteins. In our manuscript, we have now revised the relevant statements in the Results and Discussion sections to clarify that the observed associations are consistent with direct interactions predicted by AlphaFold2, but cannot alone establish direct binding. We have also tempered our terminology—substituting phrases such as “direct interaction” with “physical association consistent with direct binding,” where appropriate.

(4) Based on prior studies it would be expected that the large majority of both septins and exocyst subunits are present in cells and extracts as stoichiometric complexes. Thus, one would expect any septin-exocyst interaction to yield associations detectable with multiple subunits, yet co-IPs were not detected in some combinations. It is therefore unclear whether the interactions reflect associations between fully-formed functional complexes or perhaps between transient folding intermediates.

We thank the reviewer for this thoughtful observation. We agree that both septins and exocyst subunits are generally understood to exist in cells as stable, stoichiometric complexes, and that interactions between fully assembled complexes might be expected to yield co-immunoprecipitation signals involving multiple subunits from each complex. However, it was also found that >50% of septins Spn1 and Spn4 are in the cytoplasm even during cytokinesis when the septin double rings are formed (Table 1 of Wu and Pollard, Science 2005, PMID: 16224022). Thus, it is possible that there are pools of free septin and exocyst subunits in the cytoplasm, which were detected in our Co-IP assays.

In our experiments, we observed selective co-IP signals between certain septin and exocyst subunits, while other combinations did not yield detectable interactions. We believe these findings could reflect several other possibilities besides the possible interactions among the free subunits in the cytoplasm:

(1) Some interactions may only be strong enough between specific subunits at exposed interfaces under the Co-IP conditions, rather than through wholesome complex–complex interactions;

(2) The detergent and/or salt conditions used in our co-IPs may disrupt labile complex interfaces or partially dissociate multimeric assemblies.

To address this concern, we now include in the Discussion a paragraph highlighting the possibility that some of the observed interactions may not reflect binding between fully assembled, functional complexes. Notably, most detected interactions pairs are consistent with the AlphaFold predictions, which suggest specific subunit interfaces may be responsible for mediating contact. While we cannot fully resolve whether septins engage with the whole exocyst complex versus selected subunits, our combined data supports a model that septins scaffold or spatially regulate the exocyst localization at the division site, potentially through dynamic and multivalent interactions. We now explicitly state this more cautious interpretation in the revised manuscript. Future biochemical studies using native complex purifications, cross-linking mass spectrometry, or in vitro reconstitution with fully assembled septin and exocyst complexes, or in vivo FRET assays will be essential to clarify whether the interactions we observe occur between intact assemblies or intermediate forms.

**Reviewer #1 (Recommendations for the Authors):**
A major finding from the manuscript is the description of physical interaction of septin subunits with exocyst subunits. The analysis starts from Alphafold2 predictions, shown in Figures 3 and S3. However, some of the most useful metrics of Alphafold, the PAE plot and the pTM and ipTM values, are not provided. It is thus very difficult to estimate the value of the predicted structures (which are also obscured by all side chains). The power of a predicted structure is that it suggests binding interfaces, which is not explored here. At the very least, it would not be difficult to examine whether the proposed binding interfaces are free in the septin filaments and octameric exocyst complex.

Please also see response to reviewer #1 (Public Review).

We thank the reviewer for these very helpful suggestions. We agree that inclusion of AlphaFold2 model confidence metrics—specifically the Predicted Aligned Error (PAE) plots, as well as pTM and ipTM values—is essential for evaluating the reliability of the predicted septin–exocyst interfaces.

In the revised manuscript, we have now included the PAE plots (Figure 3 and Supplementary S3) and summarizes the pTM scores for each predicted septin–exocyst subunit pair. We also provide a short description of these metrics in the figure legend to help guide interpretation. The old Alphafold2 version (alphafold2advanced) that we used doesn’t give iPTM score, so are not included. However, according to our methodology, we only counted the interacting residues which have pLDDT scores >50%, predicting the resulting iPTM score should not be very weak.

In addition, we have updated Figures 3 and S3 to show simplified ribbon diagrams of the interface regions, with side chains hidden by default and selectively displayed only at predicted interaction hotspots. This improves structural clarity and makes the interface regions easier to interpret. We mentioned in the Discussion that the preliminary studies show that the predicted interacting interfaces of Sec15 and Sec5 with septin subunits are accessible for interaction in the whole exocyst complex. The new Figure Supplement S4 and S5 and Videos 4-7 now show the interface residues of both the exocyst and septins that are involved in the interactions.

Two further points on the interaction:The 2H interaction data is not very convincing. The insets showing beta-gal assays do not look very different from the negative control (compare for instance in panel 4E the Sec15BD alone, last column, with the Sec15-BD in combination with Spn4-AD, third column: roughly same color), which suggests it is mostly driven by autoactivation of Sec15-BD. Providing growth information in addition to beta-gal may be helpful.

We appreciate the reviewer’s close evaluation of the yeast two-hybrid (Y2H) assay data, and we agree that the signals observed in the Spn4–Sec15 combination is indeed weak. Unfortunately, we did not perform growth assays. However, we would like to clarify that this is consistent with the nature of the interactions that we are investigating. The interaction between individual septin and exocyst subunits is not strong and/or transient as supported by the weak interactions by Co-IP experiments. Given the exocyst only tethers/docks vesicles on the plasma membrane for tens of seconds before vesicle fusion, the multivalent interactions between septins and the exocyst should be very dynamic and not be too strong.

As evidenced by our Co-IP experiments and multivalent interactions predicted by Alphafold2, the interaction between Spn4 and Sec15 is detectable but weak, suggesting that this may be a low-affinity or transient interaction. Given that Y2H assays have known limitations in detecting such low-affinity interactions—especially those that depend on conformational context or are not optimal in the yeast nucleus—it is perhaps not surprising that the X-gal color development is subtle. These limitations of the Y2H system have been well-documented (e.g., Braun et al., 2009; Vidal & Fields, 2014), particularly for interactions with affinities in the micromolar range or those requiring conformational specificity. Therefore, the weak signal observed is in line with expectations for a lowaffinity, transient interaction such as between Spn4 and Sec15.

Vidal, M. and Fields, S., 2014. The yeast two-hybrid assay: still finding connections after 25 years. Nature methods, 11(12), pp.1203-1206.

Braun, P., Tasan, M., Dreze, M., Barrios-Rodiles, M., Lemmens, I., Yu, H., Sahalie, J.M., Murray, R.R., Roncari, L., De Smet, A.S. and Venkatesan, K., 2009. An experimentally derived confidence score for binary protein-protein interactions. Nature methods, 6(1), pp.91-97.

In the coIP experiments, I am confused by the presence of tubulin signal in some of the IPs. For instance, in Fig 4B, but not 4D, where the same Sec15-GFP is immunoprecipitated. There is also a signal in 4C but not 4A. This needs to be clarified.

The presence of tubulin in some immunoprecipitates is not unexpected, particularly in experiments involving cytoskeleton-associated proteins such as septins and exocyst subunits. The occasional presence of tubulin in our co-IP samples is consistent with well-documented reports showing tubulin as a frequent non-specific co-purifying protein, particularly under native lysis conditions used to preserve large complexes (Vega and Hsu, 2003; Gavin et al., 2006; Mellacheruvu et al., 2013; Hein et al., 2015). The CRAPome database and quantitative interactomics studies highlight tubulin as one of the most common background proteins in affinity-based workflows. Importantly, tubulin was used as a loading control but not as a marker for interaction in our study, and its variable presence does not reflect a specific interaction with Sec15-GFP or other bait proteins, and we have clarified this point in the revised figure legend.

Gavin, A.C., Aloy, P., Grandi, P., Krause, R., Boesche, M., Marzioch, M., Rau, C., Jensen, L.J., Bastuck, S., Dümpelfeld, B. and Edelmann, A., 2006. Proteome survey reveals modularity of the yeast cell machinery. Nature, 440(7084), pp.631-636.

Mellacheruvu, D., Wright, Z., Couzens, A.L., Lambert, J.P., St-Denis, N.A., Li, T., Miteva, Y.V., Hauri, S., Sardiu, M.E., Low, T.Y. and Halim, V.A., 2013. The CRAPome: a contaminant repository for affinity purification–mass spectrometry data. Nature methods, 10(8), pp.730736.

Hein, M.Y., Hubner, N.C., Poser, I., Cox, J., Nagaraj, N., Toyoda, Y., Gak, I.A., Weisswange, I., Mansfeld, J., Buchholz, F. and Hyman, A.A., 2015. A human interactome in three quantitative dimensions organized by stoichiometries and abundances. Cell, 163(3), pp.712-723.

Vega, I.E., Hsu, S.C. 2003. The septin protein Nedd5 associates with both the exocyst complex and microtubules and disruption of its GTPase activity promotes aberrant neurite sprouting in PC12 cells. Neuroreport, 14, pp.31-37.

Regarding the localization of Ypt3 and Syb1 in WT and spn1∆ in Figure 6C-D and Bgs1 in Figure 7A, it would help to add a contractile ring marker to be able to match the timing of cytokinesis between WT and mutants and ensure that cells of same stage are compared (and add some quantification for Ypt3). In fact, in Figure 7A, next to the cells being pointed at, there are very similar localizations of Bgs1 in WT and spn1∆ at the rim of the ingressing septum, which makes me wonder how the quantified cells were chosen.

For localizations and quantifications of Eng1, Ypt3, Syb1, and Bgs1 shown in Figures 6 and 7, cells with a closed septum (at or after the end of contractile-ring constriction) were quantified or highlighted. To quantify their fluorescence intensity at the division site using line scan, the line width used was 3 pixels. For Syb1 (Figure 6D), we quantified cells at the end of ring constriction (when Rlc1-tdTomato constricted to a dot) in the middle focal plane. The exact same lines were drawn in both Rlc1 and Syb1 channels. The center of line scan was defined as the pixel with the brightest Rlc1 value. All data were aligned by the center and plotted. For Bgs1 (Figure 7A), we quantified the cells that Rlc1 signal had disappeared from the division site. The line was drawn in the Bgs1 channel in the middle focal plane. The center of line scan was defined as the pixel with the brightest Bgs1 value. All data were aligned by the center and plotted. These details were added to the Materials and Methods.

Finally, the manuscript would benefit from some figure reorganization/compaction. Unless work on the binding interfaces is added, Figure 3 and S3 could be removed and summarized by providing the pTM and ipTM values of the predicted interactions. Figure 5 could be combined with Figure 2, as it is essentially a repeat with additional exocyst subunits.

Because the binding interfaces are added, we keep the original Figures 3 and S3. The experiments in Figure 5 could not be performed before the interaction tests between septins and the exocyst. Thus, to aid the flow of the story, we keep Figures 2 and 5 separated.

Minor comments:The last sentence of the first paragraph of the results does not make much sense at this point of the paper. After the first paragraph, there is no evidence that colocalization would be required for proper function.

We agree that the sentence in question may have overstated the functional implications of colocalization too early in the Results section, before presenting supporting evidence. Our intention was to introduce the hypothesis that spatial proximity between septins and exocyst subunits may be relevant for their coordination during cytokinesis, which we examine in later figures. We have revised the sentence to more accurately reflect the observational nature of the data at this stage in the manuscript as below:

"These observations suggest the spatial proximity between septins and the exocyst during certain stage of cytokinesis, raising the possibility of their functional coordination, which we would further investigate below."

What is the indicated n in Figure 6B? Number of cells?

Yes, the n in Figure 6B refers to the thin sections of electron microscopy quantified in the analysis. We have now updated the figure legend to explicitly state this for clarity.

The causal inference made between the alteration of Exocyst localization in septin mutants and the thicker septum is possible, but by no means certain. It should be phrased more cautiously.

We agree that our original phrasing may have overstated the causal relationship between altered exocyst localization in septin mutants and septum thickening. Our data supports a correlation between these phenotypes, but additional experiments would be required to establish direct causality.

To reflect this, we have revised the relevant sentence in the Discussion to read:

“The modest accumulation of vesicles and vesicle cargos at the division site is one of the reasons for the increased thickness of the division septum in septin mutants. It is more likely that the misplaced exocyst can still tether vesicles along the division plane without septins. Due to the lack of the glucanase Eng1 at the rim of the division plane in septin mutants, daughter-cell separation is delayed and then cells continue to thicken the septum.”

**Reviewer #2 (Recommendations for the Authors):**
(1) In the display of the AlphaFold Model for the interactions (Figure 3 and Supplemental Figure 3) it is difficult to identify which subunits are where. Residue numbers and subunits should be labeled and only side chains important for the interactions should be present in the model.

We appreciate this valuable suggestion. We agree that clearer visual labeling is essential for interpreting the predicted interactions and have revised Figures 3 and S3 accordingly to improve readability and emphasize key structural features.

Specifically, we have:

• Labeled each subunit with its name and color-coded consistently across panels.

• Annotated key interface residues with residue numbers directly in the figure.

• Removed non-interacting side chains to declutter the model and highlight only those involved in predicted interactions as well as expanded the figure legend for explanation.

(2) In Table 1 the column label "Genetic Interaction at 25C" is confusing when synthetic growth defects are shown with a "plus". Rather this column could be labeled "Growth of double mutants at 25C" and then designate the relative growth rate observed at 25C as in Table 2. Designating a negative effect on growth with a plus is confusing.

Thanks for the thoughtful suggestions. We have made the suggested changes by deleting the last column so that Tables 1 and 2 are consistent.

(3) In Figure 4, why is tubulin being co-immunoprecipitated in two of the four anti-GFP IPs? Are the IPs dirty and if so why does it vary between the four experiments? If they are dirty can the non-specific tubulin be removed by additional washes with IP buffer or conversely is it necessary to do minimal washes in order to detect the exocyst-septin interaction by coIP? A comment on this would be helpful.

The presence of tubulin in some immunoprecipitates is not unexpected, particularly in experiments involving cytoskeleton-associated proteins such as septins and exocyst subunits. The occasional presence of tubulin in our co-IP samples is consistent with welldocumented reports showing tubulin as a frequent non-specific co-purifying protein, particularly under native lysis conditions used to preserve large complexes (Vega and Hsu, 2003; Gavin et al., 2006; Mellacheruvu et al., 2013; Hein et al., 2015). The CRAPome database and quantitative interactomics studies highlight tubulin as one of the most common background proteins in affinity-based workflows. Importantly, tubulin was used as a loading control but not marker for interaction in our study, and its variable presence does not reflect a specific interaction with Sec15-GFP or other bait proteins, and we have clarified this point in the revised figure legend.

Gavin, A.C., Aloy, P., Grandi, P., Krause, R., Boesche, M., Marzioch, M., Rau, C., Jensen, L.J., Bastuck, S., Dümpelfeld, B. and Edelmann, A., 2006. Proteome survey reveals modularity of the yeast cell machinery. Nature, 440(7084), pp.631-636.

Mellacheruvu, D., Wright, Z., Couzens, A.L., Lambert, J.P., St-Denis, N.A., Li, T., Miteva, Y.V., Hauri, S., Sardiu, M.E., Low, T.Y. and Halim, V.A., 2013. The CRAPome: a contaminant repository for affinity purification–mass spectrometry data. Nature methods, 10(8), pp.730736.

Hein, M.Y., Hubner, N.C., Poser, I., Cox, J., Nagaraj, N., Toyoda, Y., Gak, I.A., Weisswange, I., Mansfeld, J., Buchholz, F. and Hyman, A.A., 2015. A human interactome in three quantitative dimensions organized by stoichiometries and abundances. Cell, 163(3), pp.712-723.

Vega, I.E., Hsu, S.C. 2003. The septin protein Nedd5 associates with both the exocyst complex and microtubules and disruption of its GTPase activity promotes aberrant neurite sprouting in PC12 cells. Neuroreport, 14, pp.31-37.

In response to the second part of reviewer’s comment, we washed the pulldown product for 5 times each time with 1 ml IP buffer at 4ºC. We used this standard protocol for all the Co-IP experiments to detect the interaction between different septin-exocyst subunits. So, we are not sure if and how more washes or more stringent buffer conditions can interfere with detection of the interactions.

**Reviewer #3 (Recommendations for the Authors):**
In addition to the issues noted in the public review, there were some confusing findings and references to previous literature that merit further consideration or discussion:• The current gold standard for validating Alphafold predictions involves making targeted mutants suggested by the structural predictions. The absence of any such validation weakens the conclusions significantly.

We agree that the targeted mutagenesis based on AlphaFold2-predicted interaction interfaces represents a powerful approach to experimentally validate the in silico models. While we did not pursue structure-guided mutagenesis in this study, our goal was to identify putative interactions between septin and exocyst subunits as a foundation for future functional work. Our current conclusions are intentionally limited to proposing putative interfaces, supported by co-immunoprecipitation and genetic interaction data.

We recognize that direct validation of specific contact residues would significantly strengthen the model. Accordingly, we have revised the Discussion to explicitly state this limitation and to note that structure-based mutagenesis will be an important next step to test the functional relevance of predicted interactions. We have added the following statement:

“Future studies are needed to refine the residues involved in the interactions because the predicted interacting residues from AlphaFold are too numerous. However, it is encouraging that most of the predicted interacting residues are clustered in several surface patches. Experimental validation through targeted mutagenesis is an important next step.”

• Much of the writing appears to imply that differences in mutant phenotypes indicate differences in septin (or exocyst) subunit behaviors/functions. However, my reading of the work in budding yeast is that such differences reflect the partial functionality that can be conferred by aberrant partial septin complexes that assemble and may polymerize in mutants lacking different subunits. In this view, which is supported by data showing that essentially all septins are in stoichiometric octameric complexes in cells, the wild-type functions are all mediated by the full complex. Similarly, the separate exocyst subunit localizations based on tagged Sec3 (Finger et al) were not supported by later work from the Brennwald lab with untagged Sec3, and the idea that different exocyst subunits may function separately from the full complex has very limited support in yeast. I would suggest that the text be edited to better reflect the literature, or that different views be better justified.

Thanks for the suggestions. We have revised the text accordingly.

• The comprehensive set of Alphafold2 predictions is a major strength of the paper, but it is unclear to this reader whether the multiple predicted interactions truly reflect multivalent multimode interactions or whether many (most?) predictions would not be consistent with interactions between full complexes and may not indicate physiological interactions. Better discussion of these issues is needed to interpret the findings.

We appreciate the reviewer’s suggestion to use structural prediction to further assess interaction plausibility. We have now employed the full *Saccharomyces cerevisiae* exocyst complex (with 4.4 Å resolution) published by the Guo group to examine the interfaces of septins and the exocyst interactions, assuming that the *S. pombe* exocyst has the similar structure. We mapped predicted contact residues onto the predicted structure. Most predicted interfaces (86% for the exocyst and 86-96% for septins) appear to be located on accessible surfaces in the assembled complexes (Figure supplement S4, S5, videos 4 - video 7), suggesting that these interactions are sterically plausible. We have added this important caveat to the text of the revised manuscript highlighting the interface accessibility within the assembled complexes. We appreciate the reviewer’s insight, which helped us strengthen the interpretation and limitations of the AlphaFold-based analysis.

• Some but not all co-IP blots appear to show tubulin (negative control) coming down with the GFP pull-downs. Why is that, and what does it imply for the reliability of the co-IP protocol?

The presence of tubulin in some immunoprecipitates is not unexpected, particularly in experiments involving cytoskeleton-associated proteins such as septins and exocyst subunits. The occasional presence of tubulin in our co-IP samples is consistent with welldocumented reports showing tubulin as a frequent non-specific co-purifying protein, particularly under native lysis conditions used to preserve large complexes (Vega and Hsu, 2003; Gavin et al., 2006; Mellacheruvu et al., 2013; Hein et al., 2015). The CRAPome database and quantitative interactomics studies highlight tubulin as one of the most common background proteins in affinity-based workflows. Importantly, tubulin was used as a loading control but not a marker for interaction in our study, and its variable presence does not reflect a specific interaction with Sec15-GFP or other bait proteins, and we have clarified this point in the revised figure legend.

Gavin, A.C., Aloy, P., Grandi, P., Krause, R., Boesche, M., Marzioch, M., Rau, C., Jensen, L.J., Bastuck, S., Dümpelfeld, B. and Edelmann, A., 2006. Proteome survey reveals modularity of the yeast cell machinery. Nature, 440(7084), pp.631-636.

Mellacheruvu, D., Wright, Z., Couzens, A.L., Lambert, J.P., St-Denis, N.A., Li, T., Miteva, Y.V., Hauri, S., Sardiu, M.E., Low, T.Y. and Halim, V.A., 2013. The CRAPome: a contaminant repository for affinity purification–mass spectrometry data. Nature methods, 10(8), pp.730736.

Hein, M.Y., Hubner, N.C., Poser, I., Cox, J., Nagaraj, N., Toyoda, Y., Gak, I.A., Weisswange, I., Mansfeld, J., Buchholz, F. and Hyman, A.A., 2015. A human interactome in three quantitative dimensions organized by stoichiometries and abundances. Cell, 163(3), pp.712-723.

Vega, I.E., Hsu, S.C. 2003. The septin protein Nedd5 associates with both the exocyst complex and microtubules and disruption of its GTPase activity promotes aberrant neurite sprouting in PC12 cells. Neuroreport, 14, pp.31-37.

• Why were two different protocols used for different yeast-two-hybrid analyses?

The purpose of using two protocols was to test which protocol is more reliable and sensitive.

• The different genetic interactions between septin and exocyst mutants when combined with TRAPP-II mutants merits further discussion: might the difference reflect relocation of exocyst from rim to center in septin mutants versus inactivation of exocyst in exocyst mutants?

We appreciate this insightful comment and agree that this distinction is likely meaningful. The reviewer correctly notes that septin mutants may not abolish exocyst function but rather cause its spatial mislocalization: from the rim to the center of the division site, whereas the exocyst mutants likely result in partial or complete loss of vesicle tethering activity at the plasma membrane.

To address this important nuance, we have expanded the Discussion as follows:

“The genetic interactions between mutations in the exocyst and septins when combined with TRAPP-II mutants may reflect fundamentally different consequences for compromising the exocyst function (Tables 1 and 2). In septin mutants, the exocyst complex still localizes to the division site but is mispositioned from the rim to the center of the division plane. This mislocalization allows partial retention of exocyst function, leading to very mild synthetic or additive defects when combined with compromised TRAPP-II trafficking and tethering. In contrast, in exocyst subunit mutants, the exocyst becomes partial or non-functional, resulting in a more severe loss of exocyst activity. These differing consequences could explain the qualitative differences in genetic interactions observed with TRAPP-II mutants (Tables 1 and 2). Thus, septins and the exocyst also work in different genetic pathways for certain functions in fission yeast cytokinesis.”

• The vesicle accumulation in septin mutants was quite modest. Does that imply that most vesicles are still fusing in the septum? Further discussion would be beneficial to understand what the authors think this means.

We thank the reviewer for this important point. We agree that the modest vesicle accumulation observed in septin mutants suggests that a significant proportion of vesicles continue to successfully fuse at the division site, even in the absence of fully functional septin structures.

We now discuss this in greater detail in the revised manuscript:

“The relatively modest vesicle accumulation in septin mutants suggests that septins are not absolutely required for vesicle tethering or fusion per se at the division site. Instead, septins primarily function to spatially organize the targeting sites of exocyst-directed vesicles by stabilizing the localization of the exocyst at the rim of the cleavage furrow. In septin mutants, mislocalization of the exocyst reduces the spatial precision of membrane insertion but still permits vesicle tethering and fusion, albeit in a less controlled manner. Thus, septins likely play a modulatory rather than essential role in exocytic vesicle delivery during cytokinesis. This interpretation aligns with our localization and genetic interaction data, which indicates that septins act as scaffolds to optimize secretion geometry, rather than as core components of the fusion machinery.”

• It was unclear to this reader why relocation of some exocyst complexes from the rim to the center of the septal region would lead to dramatic thickening of the septum. Further discussion would be beneficial to understand what the authors think this means.

The modest accumulation of vesicles and vesicle cargos at the division site is one of the reasons for the increased thickness of the division septum in septin mutants. It is more likely that the misplaced exocyst can still tether vesicles along the division plane without septins. Because of the lack of glucanase Eng1 at the rim of the division plane in septin mutants, daughter-cell separation is delayed and then cells continue to thicken the septum. We have added these points to the Discussion.